



# EC-Earth3-AerChem, a global climate model with interactive aerosols and atmospheric chemistry participating in CMIP6

Twan van Noije[1], Tommi Bergman[1,2], Philippe Le Sager[1], Declan O'Donnell[2], Risto Makkonen[2,3], María Gonçalves-Ageitos[4,5], Ralf Döscher[6], Uwe Fladrich[6], Jost von Hardenberg[7], Jukka-Pekka Keskinen[2,3], Hannele Korhonen[2], Anton Laakso[8], Stelios Myriokefalitakis[9], Pirkka Ollinaho[2], Carlos Pérez García-Pando[4,10], Thomas Reerink[1], Roland Schrödner[11,*], Klaus Wyser[6], and Shuting Yang[12]

[1]Royal Netherlands Meteorological Institute, De Bilt, Netherlands
[2]Climate System Research, Finnish Meteorological Institute, Helsinki, Finland
[3]Institute for Atmospheric and Earth System Research, Faculty of Science, University of Helsinki, Helsinki, Finland
[4]Barcelona Supercomputing Center, Barcelona, Spain
[5]Technical University of Catalonia, Barcelona, Spain
[6]Swedish Meteorological and Hydrological Institute, Norrköping, Sweden
[7]Institute of Atmospheric Sciences and Climate, National Research Council, Turin, Italy
[8]Atmospheric Research Centre of Eastern Finland, Finnish Meteorological Institute, Kuopio, Finland
[9]Institute for Environmental Research and Sustainable Development, National Observatory of Athens, Athens, Greece
[10]ICREA, Catalan Institution for Research and Advanced Studies, Barcelona, Spain
[11]Centre for Environmental and Climate Research, Lund University, Lund, Sweden
[12]Danish Meteorological Institute, Copenhagen, Denmark
[*]now at: Leibniz Institute for Tropospheric Research, Leipzig, Germany

*Correspondence to*: Twan van Noije (noije@knmi.nl)

**Abstract.** This paper documents the global climate model EC-Earth3-AerChem, one of the members of the EC-Earth3 family of models participating in the Coupled Model Intercomparison Project phase 6 (CMIP6). EC-Earth3-AerChem has interactive aerosols and atmospheric chemistry and contributes to the Aerosols and Chemistry Model Intercomparison Project (AerChemMIP). In this paper, we give an overview of the model and describe in detail how it differs from the other EC-Earth3 configurations, and what the new features are compared to the previously documented version of the model (EC-Earth 2.4). We explain how the model was tuned and spun up under pre-industrial conditions and characterize the model's general performance on the basis of a selection of coupled simulations conducted for CMIP6. The mean energy imbalance at the top of the atmosphere in the pre-industrial control simulation is -0.10 ± 0.25 W m$^{-2}$ and shows no significant drift. The corresponding mean global surface air temperature is 14.05 ± 0.16 °C, with a small drift of -0.075 ± 0.009 °C per century. The model's effective equilibrium climate sensitivity is estimated at 3.9 °C and its transient climate response at 2.1 °C. The CMIP6 historical simulation displays spurious interdecadal variability in Northern Hemisphere temperatures, resulting in a large spread among ensemble members and a tendency to underestimate observed annual surface temperature anomalies from the early 20th century onwards. The observed warming of the Southern Hemisphere is well reproduced by the model. Compared to the ERA5 reanalysis of the European Centre for Medium-Range Weather Forecasts, the ensemble mean surface air temperature climatology for 1995–2014 has an average bias of -0.86 ± 0.35 °C in the Northern Hemisphere and 1.29 ± 0.05 °C in the Southern Hemisphere. The Southern Hemisphere warm bias is largely caused by errors in shortwave cloud radiative effects over the Southern Ocean, a deficiency of many climate models. Changes in the emissions of near-term climate forcers (NTCFs) have significant climate effects from the 20th century onwards. For the SSP3-7.0 shared socio-economic pathway, the model gives a global warming at the end of the 21st century (2091–2100) of 4.9 °C above the pre-industrial mean. A 0.5 °C stronger warming is obtained for the AerChemMIP scenario with reduced emissions of NTCFs. With concurrent reductions of future methane concentrations, the warming is projected to be reduced by 0.5 °C.



# 1 Introduction

EC-Earth is a global climate and Earth system model developed by a European consortium of meteorological services, research institutes, and high-performance computing centres (Hazeleger et al., 2010; 2012). Activities in recent years were dedicated to the development of the third generation of the model and participation in the Coupled Model Intercomparison Project phase 6 (CMIP6; Eyring et al., 2016). The basic CMIP6 configuration of the model (EC-Earth version 3.3.1.1, hereafter referred to as EC-Earth3) consists of an atmospheric general circulation model (GCM) based on cycle 36r4 of the Integrated Forecasting System (IFS) from the European Centre for Medium-Range Weather Forecasts (ECMWF), coupled to the NEMO-LIM3 global ocean–sea ice model from the Nucleus for European Modelling of the Ocean (NEMO) release 3.6 (Rousset et al., 2015). Other CMIP6 configurations include additional modules for simulating dynamic vegetation, the carbon cycle, aerosols and atmospheric chemistry, or the Greenland ice sheet. Low- and high-resolution configurations have also been developed. An overview of the different configurations is given by Döscher et al. (in preparation). Specific model configurations will be documented in separate publications. This paper documents the configuration with interactive aerosols and atmospheric chemistry (EC-Earth-AerChem version 3.3.3, hereafter EC-Earth3-AerChem). It is with this configuration that the EC-Earth consortium participates in the Aerosols and Chemistry Model Intercomparison Project (AerChemMIP; Collins et al., 2017).

The distinguishing feature of EC-Earth3-AerChem compared to the other EC-Earth configurations applied in CMIP6, is that it simulates tropospheric aerosols and the reactive greenhouse gases methane and ozone. In all other configurations these are prescribed as described by Döscher et al. (in preparation). Although methane and stratospheric ozone are not fully prescribed in EC-Earth3-AerChem, they are constrained by the CMIP6 forcing data sets of Meinshausen et al. (2017; 2020) and Checa-Garcia et al. (2018), respectively. As a result, the main differences between EC-Earth3-AerChem and the other CMIP6 configurations of EC-Earth are related to tropospheric aerosols and tropospheric and lower-stratospheric ozone, and how they interact with the climate system.

In this paper we describe the model and present first results from CMIP6 simulations. The remainder of the paper is structured as follows. The model description is given in Sect. 2. In this section we subsequently describe the main general characteristics of the model as well as the treatment of aerosols and their interactions with radiation and clouds, atmospheric chemistry and chemical boundary conditions, and anthropogenic and natural emissions, and finally some relevant technical and numerical aspects of the model. The applied tuning and spin-up procedures are outlined in Sect. 3. Results from CMIP6 simulations are presented in Sect. 4. The analysis presented in this section focuses on the effective equilibrium climate sensitivity and the transient climate response, the net energy imbalance in the simulations, and the long-term evolution and present-day climatology of surface air temperatures. Finally, we end the paper with a discussion and conclusions in Sect. 5.

# 2 Model description

## 2.1 General

EC-Earth3-AerChem is essentially EC-Earth3 (Döscher et al., in preparation) extended with an additional component to simulate aerosols and atmospheric chemistry. The atmospheric GCM is based on IFS cycle 36r4, which includes the land surface model H-TESSEL (revised hydrology version of TESSEL, the Tiled ECMWF Scheme for Surface Exchanges over Land; Balsamo et al., 2009). Its horizontal resolution is T$_l$255 (triangular truncation at wavenumber 255 in spectral space with a linear N128 reduced



Gaussian grid, corresponding to a spacing of about 80 km). The atmospheric grid consists of 91 layers in the vertical direction and

has a model top at 0.01 hPa. The time step applied in IFS is 45 min.

The McRad radiation package of cycle 36r4 consists of a shortwave (SW) and longwave (LW) radiation scheme based on the Rapid Radiative Transfer Model for General Circulation Models (RRTMG), and uses the Monte Carlo Independent Column Approximation (McICA) to treat the radiative transfer in clouds (Morcrette et al., 2008). Clouds and large-scale precipitation are

described by prognostic equations for cloud liquid water, cloud ice, rain, snow and a grid box fractional cloud cover.

Compared to the original model from IFS cycle 36r4, several adjustments and updates have been made in EC-Earth (Döscher et al., in preparation). These include the application of global mass fixers for dry air and humidity (Diamantakis and Flemming, 2014), a resolution-dependent parameterisation of non-orographic gravity wave drag (Davini et al., 2017), and a diagnostic

convective closure, which is dependent on the convective available potential energy (Bechtold et al., 2014). Moreover, the Abdul-Razzak and Ghan (2000) cloud activation scheme has been introduced, and a dependence of the autoconversion efficiency on the cloud droplet number concentration (CDNC) has been added, following Rotstayn and Penner (2001). Details about the representation of aerosol-cloud interactions in EC-Earth3-AerChem are given in Sect. 2.2.

Moreover, CMIP6 forcings have been introduced. For EC-Earth3-AerChem, the CMIP6 forcings prescribed in IFS are the solar forcing (Matthes et al., 2017), well-mixed greenhouse gas concentrations ($CO_2$, $N_2O$, CFC-12, and CFC-11 equivalents; Meinshausen et al., 2017; 2020), and stratospheric aerosol radiative properties. Vegetation fields consistent with the CMIP6 land use forcing data sets (Hurtt et al., 2020), which have been produced using a model configuration with dynamic vegetation (EC-Earth3-Veg), replace the climatological input fields applied in the standard IFS model. The corresponding surface albedo of the

soil and vegetation is calculated as described by Döscher et al. (in preparation).

The ocean GCM is based on NEMO-LIM3 release 3.6 (Rousset et al., 2015), which consists of the Océan Parallélisé (OPA) ocean dynamics and thermodynamics model (Madec and the NEMO team, 2015) and the Louvain-la-Neuve sea ice model version 3 (LIM3; Vancoppenolle et al., 2009). Its horizontal grid is the tripolar ORCA1 grid, which has a resolution of approximately 1°

with meridional refinement down to 1/3° in the tropics (Madec and Imbard, 1996; Hewitt et al., 2011). The ocean grid consists of 75 layers. The time step applied in NEMO is 45 min. Compared to the reference version from the release, a few modifications have been made in EC-Earth, as described by Döscher et al. (in preparation): turbulent kinetic energy is not allowed to penetrate below the ocean mixed layer, the strength of the Langmuir cell circulations has been increased, and the thermal conductivity of snow on sea ice has been slightly reduced.


IFS and NEMO are coupled by exchanging fields via OASIS3-MCT version 3.0 (Craig et al., 2017), a new version of the OASIS3 (Ocean Atmosphere Sea Ice Soil version 3) coupler interfaced with the Model Coupling Toolkit (MCT) from the Argonne National Laboratory. The discharge of continental freshwater into the oceans is described using a runoff mapper that instantaneously relocates the runoff from the interior of 66 major global drainage basins to the coast where it enters the ocean as freshwater. In a

similar way, accumulated snow from the interior of the continents is removed and sent to the ocean as ice ("calving") to prevent the snow from piling up where the temperature is low. The corresponding fields are passed from IFS to NEMO via the runoff mapper, which is coupled to the GCMs through OASIS.



In atmosphere-only simulations, NEMO and the runoff mapper are replaced by an interface (called AMIP reader in EC-Earth) that
reads monthly or daily sea surface temperature and sea-ice concentration fields from a set of input files, applies temporal
interpolation if needed, and sends daily fields to IFS via OASIS.

The setup of the couplings between IFS, NEMO and the runoff mapper or between IFS and the AMIP reader is the same in EC-Earth3-AerChem as in EC-Earth3. Also, the parameter settings adopted in these components are identical in both configurations,
with the exception of three atmospheric parameters that have been slightly retuned (see Sect. 3).

Aerosols and atmospheric chemistry are simulated with the Tracer Model version 5 (TM5), specifically release 3.0 of the massively parallel version of TM5 (TM5-mp 3.0). It runs at a horizontal resolution of $3° \times 2°$ (longitude × latitude) with 34 layers in the vertical direction. By origin, TM5 is a standalone atmospheric chemistry and transport model (CTM), which is driven by offline
meteorological and surface fields (Krol et al., 2005; Huijnen et al., 2010). About a decade ago, TM5 was integrated as a module coupled to IFS within EC-Earth. A description and evaluation of the TM5-IFS coupled system in the previous generation of EC-Earth (version 2.4) was given by van Noije et al. (2014). Since then, the representation of chemistry and aerosols in TM5 has been revised in many respects, and interactions with radiation and clouds have been introduced. In the remainder of this section we will briefly describe how TM5 and relevant aspects in IFS have changed compared to the system documented in van Noije et al. (2014).


### 2.2 Aerosols and their interactions with radiation and clouds

The aerosol components represented in TM5 include sulfate ($SO_4$), black carbon (BC), organic aerosols (OA), sea salt, and mineral dust. These are described by the modal aerosol microphysical scheme M7 (Vignati et al., 2004), which consists of four water-soluble modes (nucleation, Aitken, accumulation, and coarse) and three insoluble modes (Aitken, accumulation and coarse).
Particles inside the modes are assumed to be internally mixed. Each mode is described by a log-normal size distribution with a fixed geometric standard deviation. For each mode, M7 describes the evolution of the total particle number and mass of each species. The scheme accounts for new particle formation, water uptake, and ageing through coalescence and condensation.

Other aerosol components described by TM5 are ammonium ($NH_4$), nitrate ($NO_3$), methane sulfonic acid (MSA), and the
diagnostic radioactive tracer lead-210 ($^{210}Pb$). The concentrations of ammonium, nitrate and water associated with (ammonium) nitrate are determined based on equilibrium gas/particle partitioning calculations using EQSAM (Metzger et al., 2002). Ammonium nitrate is assumed to be present only in the soluble accumulation mode. MSA is produced by oxidation of dimethyl sulfide (DMS) in the gas phase and is assumed to condense instantaneously onto existing soluble accumulation-mode particles. When calculating the mass, size and optical properties of these particles, the model accounts for the presence of ammonium nitrate and its associated
water, as well as MSA.

Aerosol water uptake is calculated using a diagnostic estimate of the clear-sky relative humidity. In grid boxes where the relative humidity (RH) is lower than or equal to 90 %, the clear-sky RH is set equal to the all-sky value; where the RH exceeds 90 %, the clear-sky RH is calculated by assuming that the RH is the area weighted average of the cloudy-sky RH, which is set to 100 %, and
the clear-sky RH, and applying a minimum value of 75 %. Water uptake by sulphate and sea salt is calculated as described by Vignati et al. (2004): for internal mixtures containing sea salt, the water uptake is calculated using the ZSR method (Zdanovskii, 1948; Stokes and Robinson, 1966); in the absence of sea salt, the water uptake associated with sulphate is calculated using the





parameterization from Zeleznik (1991); black carbon, organic matter and dust do not influence the water uptake. Additional water uptake in the presence of ammonium nitrate in the soluble accumulation mode is calculated using EQSAM.


The densities of the various aerosol components are given in Table 1. The densities of black carbon and organic aerosols have been reduced from 2.0 g cm$^{-3}$ in EC-Earth 2.4 to 1.8 g cm$^{-3}$ (Bond and Bergstrom, 2006; Bond et al., 2013) and 1.3 g cm$^{-3}$ (Turpin and Lim, 2001; Cross et al., 2007; Schmid et al., 2009; Lee et al., 2010; Kuwata et al., 2012; Nakao et al., 2013), respectively. Particulate organic matter is still assumed to have a constant carbon content. It is expressed by the ratio of the total mass of OA particles to

the mass of the carbon they contain. This ratio is used to convert emissions of primary organic aerosols (POA) expressed as organic carbon (OC) mass to OA mass. A value of 1.6 is adopted for all POA sources (Turpin and Lim, 2001; Reid et al., 2005; Aiken et al., 2008). Previously, this ratio was set to 1.4 (van Noije et al., 2014; Tsigaridis et al., 2014). For the same amount of carbon emitted into the atmosphere and corresponding particle size distribution, 14 % more OA mass and 76 % more OA particles are emitted in the current model version, as a result of the reduction in the assumed particle density and carbon content of organic

aerosols. Similarly, 11 % more BC particles are emitted as a result of the reduction in the assumed particle density of black carbon.

Table 1. Physical properties for the various aerosol components included in the model. For properties that have been updated, the numbers between parentheses indicate the values used in the earlier TM5 and EC-Earth versions documented in van Noije et al. (2014).

|  | Density (g cm$^{-3}$) | Refractive index at 550 nm | Hygroscopicity parameter |
|---|---|---|---|
| Sulfate |  |  |  |
| Sulfuric acid (H$_2$SO$_4$) | 1.841 | $1.43 + 1.0 \times 10^{-8}$ i | 0.6 |
| Sodium sulfate (Na$_2$SO$_4$) | 2.68 | Not used | 0.95 |
| Sodium bisulfate (NaHSO$_4$) | 2.435 | Not used | Not used |
| Black carbon (BC) | 1.8 (2.0) | $1.85 + 0.71$ i ($1.75 + 0.44$ i) | 0.0 |
| Organic aerosols (OA) |  |  |  |
| Primary organic aerosols (POA) | 1.3 (2.0) | $1.53 + 5.5 \times 10^{-3}$ i | 0.1 |
| Secondary organic aerosols (SOA) | 1.3 (2.0) | $1.53 + 5.5 \times 10^{-3}$ i | 0.1 |
| Sea salt | 2.165 | $1.50 + 1.0 \times 10^{-8}$ i | 1.0 |
| Mineral dust | 2.65 | $1.52 + 1.1 \times 10^{-3}$ i | 0.0 |
| Ammonium nitrate | 1.73 (1.70) | $1.43 + 1.0 \times 10^{-8}$ i | 0.6 |
| MSA | 1.48 | $1.43 + 1.0 \times 10^{-8}$ i | 0.6 |
| Water | 1.0 | $1.336 + 2.5 \times 10^{-9}$ i | - |


The representation of secondary organic aerosols (SOA) has been substantially revised. A scheme has been introduced to simulate the formation of SOA in the atmosphere in a simplified way (Bergman et al., in preparation). To separately track the SOA mass in the respective modes, the original M7 framework has been extended by adding an additional transported SOA tracer in the soluble





nucleation, accumulation, Aitken, and coarse mode as well as in the insoluble Aitken mode. Consistent with the original M7 model, SOA is not produced in the insoluble accumulation and coarse modes, which consist of mineral dust only. The properties of SOA are currently assumed to be the same as for primary organic aerosols (POA; see Table 1). In the new SOA scheme, isoprene and monoterpenes, emitted by vegetation or produced by biomass burning, are oxidized by reacting with the hydroxyl radical (OH) or ozone ($O_3$). This produces, with specified yields, either extremely low-volatility organic compounds (represented by a single non-

transported tracer called ELVOC) or semi-volatile organic compounds (represented by SVOC). ELVOC can take part in new particle formation or condense onto existing particles. SVOC, on the other hand, is too volatile to contribute to new particle formation but does produce SOA via condensation. It is assumed that all produced ELVOC and SVOC are converted into SOA in a single model time step (see Sect. 2.5). Condensation of ELVOC takes place in the kinetic regime, where the rate of condensation is proportional to the available particle surface area. Thus, the amount of SOA produced by condensation of ELVOC is distributed

across the relevant modes in proportion to the total surface of the particles they contain. In contrast, consistent with equilibrium partitioning theory, the SOA production from SVOC is distributed in proportion to the total mass of OA (i.e., POA and SOA) contained in the modes. Condensation of SVOC on nucleation-mode particles is thereby neglected.

Also, the representation of new particle formation has been revised (Bergman et al., in preparation). The original M7 scheme only

accounts for particle formation through binary homogeneous nucleation of water and sulfuric acid ($H_2SO_4$) from the gas phase. The corresponding rate of nucleation and critical cluster size are calculated using a parameterization from Vehkamäki et al. (2002), which is based on classical nucleation theory. However, existing theories of binary homogeneous nucleation tend to overestimate the sensitivity to sulfuric acid concentrations, and are not able to reproduce nucleation events that take place in the planetary boundary layer, suggesting that other trace gases like organics and ammonia are also involved in the early growth process (e.g.,

Weber et al., 1996; Jung et al., 2008; Sipilä et al., 2010; Kerminen et al., 2010; Paasonen et al., 2010). To enhance the nucleation in the boundary layer, a second nucleation mechanism has been added, which describes new particle formation in the presence of sulfuric acid and low-volatility organic compounds, represented by ELVOC. Following Riccobono et al. (2014), the corresponding rate of nucleation is expressed as a two-component power law, with a quadratic dependence on the ambient vapour concentration of sulfuric acid and a linear dependence on the ELVOC concentration. The subsequent growth of the freshly formed particles to 5

nm as a result of the condensation of sulfuric acid and ELVOC is described following Kerminen and Kulmala (2002).

Table 2. Scavenging fractions for convective and stratiform in-cloud scavenging. The numbers between parentheses indicate the values used in the earlier TM5 and EC-Earth versions documented in van Noije et al. (2014). In those versions, the distinction between liquid, mixed and ice clouds is made based on the cloud liquid and ice water content, and the scavenging fraction for

mixed clouds depends on their ratio.

| | Stratiform in-cloud scavenging | | | Convective scavenging |
|---|---|---|---|---|
| | Liquid clouds | Mixed clouds | Ice clouds | |
| Soluble modes | | | | |
| Nucleation | 0.06 (0.0) | 0.06 | 0.06 (0.0) | 0.2 (1.0) |
| Aitken | 0.25 (0.0) | 0.06 | 0.06 (0.0) | 0.6 (1.0) |
| Accumulation | 0.85 (1.0) | 0.06 | 0.06 (0.2) | 0.99 (1.0) |



| | | | | |
|---|---|---|---|---|
| Coarse | 0.99 (1.0) | 0.75 | 0.06 (0.2) | 0.99 (1.0) |
| Insoluble modes | | | | |
| Aitken | 0.2 (0.0) | 0.06 | 0.06 (0.0) | 0.2 (1.0) |
| Accumulation | 0.4 (0.0) | 0.06 | 0.06 (0.0) | 0.4 (1.0) |
| Coarse | 0.4 (0.0) | 0.4 | 0.06 (0.0) | 0.4 (1.0) |
| Bulk components | 0.85 (0.7) | 0.06 | 0.0 (0.14) | 0.99 (0.99) |

Table 3. Number and mass scavenging coefficients (mm$^{-1}$) for stratiform below-cloud scavenging. The numbers between parentheses indicate the values used in the TM5 and EC-Earth versions documented in van Noije et al. (2014).

| | Number | Mass |
|---|---|---|
| Soluble and insoluble modes | | |
| Nucleation | 0.02 ($5 \times 10^{-3}$) | $2 \times 10^{-3}$ ($5 \times 10^{-3}$) |
| Aitken | $1 \times 10^{-3}$ ($2 \times 10^{-3}$) | $2 \times 10^{-4}$ ($2 \times 10^{-3}$) |
| Accumulation | $3 \times 10^{-4}$ ($8 \times 10^{-3}$) | 0.03 ($8 \times 10^{-3}$) |
| Coarse | 0.3 (1.0) | 0.7 (1.0) |
| Bulk components | $3 \times 10^{-4}$ (1.0) | 0.03 (1.0) |


Moreover, wet deposition rates describing the removal of aerosols by clouds and precipitation have been revised, as indicated in Tables 2 and 3. Scavenging of aerosols by precipitation formation in convective and stratiform clouds is described using prescribed mode-dependent scavenging fractions (Croft et al., 2010), which for convective clouds are taken from Stier et al. (2005) and for stratiform clouds from Bourgeois and Bey (2011) (see Table 2). For stratiform clouds a distinction is made between liquid, mixed and ice clouds. Here it is assumed that clouds are liquid at temperatures above 0 °C, ice below -35 °C, and mixed in between.

Below-cloud scavenging of aerosols by stratiform precipitation is described using prescribed scavenging coefficients for the particle number and mass in each mode (see Table 3). These coefficients have been estimated from results presented by Croft et al. (2009), obtained for a standard Marshall-Palmer rain droplet size distribution and a precipitation rate of 1 mm hr$^{-1}$. Because different coefficients are applied to particle number and mass, below-cloud scavenging shifts the size distributions of the modes:

the nucleation and Aitken modes are shifted to larger sizes and the accumulation and coarse modes to smaller sizes. All scavenging processes act with the same rate on the bulk aerosol components (ammonium, nitrate, MSA, and lead-210) as on the other components contained in the soluble accumulation mode. As in earlier versions of the model, in-cloud and below-cloud scavenging by stratiform clouds is reduced by delaying the subgrid-scale mixing between cloudy and cloud-free regions (see Vignati et al., 2010b). The corresponding mixing timescale has been made dependent on the horizontal resolution; at the resolution applied in

EC-Earth3-AerChem, it has been increased from 3 to 6 hours.

The calculation of the removal of aerosols by sedimentation and surface dry deposition follows the description given by aan de Brugh et al. (2011). Also here, distinct rates are applied to the particle number and mass in each mode. Compared to the version





documented by van Noije et al. (2014), dry deposition and sedimentation of ammonium, nitrate and MSA have been added. These
are now removed at the same rate as the other components contained in the soluble accumulation mode.

The aerosols simulated by TM5 are tropospheric in the sense that they mainly originate from surface emissions and sources in the
troposphere. TM5 does not include emissions from explosive volcanoes, which may reach the stratosphere when the eruption is
sufficiently strong, or chemical processes that are specifically relevant for particle formation in the stratosphere, such as the
production of sulfuric acid by the oxidation of carbonyl sulfide (COS). TM5 does simulate the transport of particles across the
tropopause and in the stratosphere, but it cannot be assumed to provide accurate information on stratospheric aerosols. For this
reason, IFS does not make use of any aerosol data from TM5 at levels above the model tropopause. Instead, radiative effects of
stratospheric aerosols are accounted for in RRTMG$_{SW}$ and RRTMG$_{LW}$ using prescribed radiative properties from CMIP6. The
treatment of stratospheric aerosols and the calculation of the tropopause level in IFS are identical to the implementation in EC-
Earth3 (see Döscher et al., in preparation).

The calculation of aerosol optical properties in TM5 is based on Mie theory (van Noije et al., 2014). The extinction, single-
scattering albedo and asymmetry factor are calculated for each mode at a number of predefined wavelength values. Spectral
refractive indices of the various aerosol components are prescribed using input tables from three different sources. For modes
consisting of internally mixed particles, effective refractive indices are calculated using volume mixing rules derived from
effective-medium theory. Sulfate, organic aerosols, sea salt, ammonium nitrate, MSA, and water are treated as homogeneous
mixtures described by the Bruggeman mixing rule. When black carbon and/or dust are present in the mix, these are treated as
inclusions in a homogeneous background medium, using the Maxwell Garnett mixing rule.

The refractive indices of sulfate and sea salt are from the OPAC package (Optical Properties of Aerosols and Clouds; Hess et al.,
1998). The values for sulfate were obtained for a solution consisting of 75 % H$_2$SO$_4$ in water. The volume occupied by sulfate is
calculated in the optics module of TM5 by assuming that all sulfate is present in the form of sulfuric acid. The refractive indices
of MSA and ammonium nitrate are assumed to be the same as for sulfate. For black carbon, the refractive index is prescribed using
the corresponding input table from OPAC but with the real and imaginary parts scaled by 1.85/1.75 and 0.71/0.44, respectively.
By applying these scale factors, the refractive index at 550 nm is changed from the OPAC value of 1.75 + 0.44 i, used in previous
versions of the model, to 1.85 + 0.71 i, i.e. the mid-range value proposed by Bond and Bergstrom (2006). The refractive indices of
organic aerosols and mineral dust are taken from the aerosol-climate model ECHAM-HAM (see Zhang et al., 2012). For OA the
values are based on OPAC; for dust the imaginary part in the visible part of the spectrum is much lower than in OPAC. The
refractive index of water is taken from Segelstein (1981). The corresponding values at 550 nm are given in Table 1.


The resulting aerosol optical properties are input to the photolysis scheme of TM5 (see Sect. 2.3) and the shortwave radiation
scheme of IFS (RRTMG$_{SW}$). Within each spectral band, the optical properties are calculated at a single wavelength value. For the
photolysis scheme, these are the 'central' wavelength values defined within the scheme. For RRTMG$_{SW}$, they are solar-weighted
band averages, with the solar spectral irradiance distribution acting as the weighting function. The corresponding wavelength
values are 257, 313, 398, 530, 697, 973, 1269, 1447, 1767, 2040, 2308, 2752, 3407, 5254 nm. For diagnostic purposes, the optical
properties are also determined at a limited number of additional wavelengths (440, 550, and 870 nm). In accordance with the
CMIP6 data request, the contributions from the stratosphere are not included in the output optical property fields.



Absorption of longwave radiation by tropospheric aerosols is included in RRTMG$_{LW}$ using a simplified approach that makes use
of pre-computed mass attenuation coefficients (MACs) from the prognostic aerosol scheme developed in IFS by Morcrette et al.
(2009). The aerosol components represented in this scheme are sulfate, black carbon, organic aerosols, sea salt and mineral dust.
For black carbon and organic aerosols, the hydrophobic and hydrophilic components are treated separately. Sea salt and dust are
represented using three size bins. For sea salt, the dry particle diameter ranges are 0.03–0.5, 0.5–5, and 5–20 μm; for dust, the
ranges are 0.03–0.55, 0.55–9, and 9–20 μm.

For each of the tracers used in the scheme of Morcrette et al. (2009), wavelength dependent MACs are specified, which are defined
as the extinction cross section per unit dry mass. Thus, the LW absorption is calculated by mapping the modal dry component
masses simulated by TM5 onto these tracers. For sea salt and dust, all of the mass contained in the accumulation modes is put into
the first bin; of the mass in the coarse modes, 22.2 % is put into the second bin and the rest into the third bin. This percentage is
based on the assumption that the third bin contains roughly 3.5 times more mass than the second (Morcrette et al., 2009). The
aerosol water simulated by TM5 is not used in this calculation. Instead, water uptake by sulfate, hydrophilic organic aerosols, and
sea salt is taken into account via a dependence of their MACs on relative humidity. For BC, the humidity independent MAC for
the hydrophobic component is applied to both the soluble and insoluble modes. The contributions of ammonium nitrate and MSA
to the LW absorption are neglected.

Aerosol activation, i.e. the formation of cloud droplets by heterogeneous nucleation on aerosols, is described following the
activation scheme from Abdul-Razzak and Ghan (2000), which was specifically developed for modal aerosol schemes like M7. It
makes use of Köhler theory to calculate the critical supersaturation and critical particle diameter for each of the relevant water-
soluble modes, assuming uniform internal mixing inside the modes. The hygroscopicity parameters adopted in the model for the
various aerosol components are given in Table 1. The scheme utilizes an approximate expression to calculate the maximum
supersaturation for an air parcel rising adiabatically at a constant vertical velocity. This allows to estimate the diameter of the
smallest activated particle per mode and consequently the number of activated particles, as a function of updraft velocity. The
cloud droplet number concentration (CDNC) follows as the total in-cloud number concentration of activated aerosols, which is
determined by averaging over an appropriate probability distribution of updraft velocities. In EC-Earth3-AerChem and other EC-
Earth3 configurations, subgrid-scale vertical velocities are described by a Gaussian distribution with mean equal to the large-scale
vertical velocity (see e.g. Morales and Nenes, 2010). The standard deviation of the distribution is set to 0.8 m s$^{-1}$. The activation is
determined as an average over the range of positive velocities. Currently, this range is sampled using 10 evenly distributed values,
varying from 0.2 to 3.8 times the standard deviation. A minimum CDNC value of 30 cm$^{-3}$ is assumed.

For radiation calculations, the effective radius of cloud droplets is determined from the cloud liquid water content provided by the
prognostic cloud scheme and the cloud droplet number concentration (CDNC) from the diagnostic cloud activation scheme,
following Martin et al. (1994) with the drizzle correction from Wood (2000). The resulting droplet effective radius depends on the
simulated aerosol number and mass concentrations, which is an expression of the first aerosol indirect or cloud albedo effect
(Twomey, 1977). To prevent unrealistic values, the effective radius is clipped to 4–30 μm.

Autoconversion of cloud droplets into rain is treated using a formulation based on Sundqvist (1978). The rate of precipitation
formation by autoconversion in stratiform clouds is calculated as

$$S = c_0 \times q \times \left(1 - exp\left[-\left(\frac{q}{q_c}\right)^2\right]\right), \tag{1}$$





where $c_0^{-1}$ is a characteristic time scale, $q$ the cloud liquid water content, and $q_c$ a critical cloud liquid water content at which the

autoconversion starts to be efficient. The calculation of $q_c$ and $c_0$ in EC-Earth3-AerChem and other EC-Earth3 configurations differs from that in IFS cycle 36r4. Instead of using fixed values of $q_c$ over land and ocean as in the original IFS model, a critical volume-mean cloud droplet radius is specified (see e.g. Rotstayn and Penner, 2001). As a result, $q_c$ is directly proportional to CDNC. The value of the critical droplet radius was determined during the tuning of EC-Earth3 and is set to 8.75 μm (see Sect. 3). An additional, weaker dependence on CDNC is introduced via the rate coefficient $c_0$ (see Wyser et al., 2020). In accordance with

other studies (e.g. Rotstayn and Penner, 2001), $c_0$ is assumed to vary as $N_d^{-1/3}$, where $N_d$ denotes CDNC. With these modifications, the precipitation formation rate in stratiform clouds is made dependent on the simulated aerosol number and mass concentrations, in such a way that the precipitation formation efficiency, $S/q$, is reduced at larger CDNC. This is an expression of the second aerosol indirect or cloud lifetime effect (Albrecht, 1989).

**2.3 Atmospheric chemistry and boundary conditions for chemical tracers**

EC-Earth3-AerChem simulates the microphysical and chemical interaction of aerosols and trace gases in the troposphere. The chemistry scheme of TM5 accounts for gas-phase, aqueous-phase and heterogeneous chemistry (van Noije et al., 2014). The gas-phase reaction scheme is a modified version of the CB05 carbon bond mechanism (Yarwood et al., 2005). A first version of the modified scheme (mCB05) was presented by Williams et al. (2013). The scheme employed in EC-Earth3-AerChem is the extended

and updated version described by Williams et al. (2017). Photolysis rates are calculated using the modified band approach from Williams et al. (2006; 2012; 2017).

When calculating photolysis rates, scattering and absorption by aerosols are accounted for based on the online simulated optical properties from within TM5 (see Sect. 2.2), without making use of the stratospheric aerosol forcing data set from CMIP6. As a

consequence, the radiative effects of large volcanic eruptions are not explicitly accounted for in the photochemistry of the model. Effects of cloud liquid water and ice particles on photolysis rates are included as described by Williams et al. (2012), but with cloud droplet effective radii calculated following the parameterization of Martin et al. (1994), using fixed values of CDNC over land and ocean (313.2 and 42.0 cm$^{-3}$, respectively, corresponding to aerosol concentrations of 900 and 40 cm$^{-3}$ for particles in the size range 0.1–3 μm diameter), with lower and upper limits set to 4 and 16 μm.


Heterogeneous chemistry is limited to the reactive uptake of dinitrogen pentoxide ($N_2O_5$) at the surface of cloud droplets, ice particles and aerosols, and the uptake of the hydroperoxyl ($HO_2$) and nitrate ($NO_3$) radicals on aerosols. In these reactions one molecule of $N_2O_5$ produces two molecules of nitric acid ($HNO_3$), one molecule of $NO_3$ produces one molecule of $HNO_3$, and two molecules of $HO_2$ produce one molecule of $H_2O_2$. These reactions are described using a first-order rate coefficient, which for a

mono-disperse distribution of spherical particles of radius $r$ is given by

$$dk(r) = \left(\frac{r}{D_g} + \frac{4}{v\gamma}\right)^{-1} dS(r), \qquad (2)$$

where $D_g$ is the gas-phase molecular diffusion coefficient of the reacting species in air, $v$ the mean molecular speed of the species in the gas phase, $\gamma$ the probability that a molecule impacting the surface undergoes reaction, and $dS(r)$ the surface area density of the particles per unit volume of air (see e.g. Jacob, 2000). The first term between the parentheses on the right-hand side of the

equation describes the uptake associated with diffusion to the particle surface, the second term describes the uptake associated with free molecular motion to the particle surface.



Following Huijnen et al. (2014), the uptake of $N_2O_5$ by heterogeneous reactions in clouds is determined from Eq. (2) by replacing the variable $r$ in the diffusion term by the effective radius $r_e$ of the droplets or ice particles, respectively. Thus, the total rate coefficient is expressed as

$$k = \left(\frac{r_e}{D_g} + \frac{4}{v\gamma}\right)^{-1} S,$$
(3)

where $S$ is the total surface area density of the liquid or ice contained in the cloud. The effective radii and surface area densities are calculated as in Williams et al. (2017). In Eq. (3) a temperature dependent reaction probability $\gamma(T)$ is used for the uptake of $N_2O_5$ on liquid water (Ammann et al., 2013), while a fixed $\gamma$ value of 0.02 is adopted for the uptake on ice particles (Crowley et al., 2010).

For aerosols the uptake tends to be limited by free molecular motion (Jacob, 2000). When performing the integration of Eq. (2) over particle size and composition, we therefore replace $\left(\frac{r}{D_g} + \frac{4}{v\gamma}\right)$ for all particles in mode $i$ by $\left(\frac{r_{g,i}}{D_g} + \frac{4}{v\gamma_i}\right)$, where $r_{g,i}$ is the geometric mean (i.e. median) radius of the mode and $\gamma_i$ the reaction probability for that mode. The total rate coefficient is thus given by

$$k = \sum_i \left(\frac{r_{g,i}}{D_g} + \frac{4}{v\gamma_i}\right)^{-1} S_i,$$
(4)

where $S_i$ is the total surface area density of the particles contained in mode $i$. For simplicity, the uptake on aerosols is described using constant values of the reaction probability, irrespective of mode or composition. For $N_2O_5$ a value of 0.02 is assumed, which corresponds to the global mean value from Evans and Jacob (2005); the reaction probabilities for $HO_2$ and $NO_3$ are set to 0.06 and 0.001, respectively, based on Abbatt et al. (2012) and Jacob (2000).

The model includes aqueous-phase reactions for the oxidation of total dissolved sulfur dioxide by dissolved hydrogen peroxide ($H_2O_2$) and ozone, depending on the acidity of the droplets (van Noije et al., 2014). The acidity calculation is done with a spatially homogeneous $CO_2$ mixing ratio, which is set equal to the annual and global mean surface value provided in the CMIP6 historical or scenario data sets (Meinshausen et al., 2017; 2020).

Other boundary conditions are applied to constrain the mixing ratios of ozone, carbon monoxide, nitric acid and methane in the stratosphere and that of methane in the lower part of the troposphere (Williams et al., 2017). These boundary conditions are applied through Newtonian relaxation (nudging) towards daily-varying zonal mean fields, obtained from monthly input data sets by linear interpolation in time. Compared to the description given by Williams et al. (2017), the boundary conditions applied for ozone and methane have been modified for CMIP6, as described below.

The mixing ratios of ozone in the stratosphere are nudged towards zonal mean fields calculated from the three-dimensional input data sets provided by CMIP6 (Checa-Garcia et al., 2018). For ozone, these consist of a climatology for the pre-industrial period and transient data sets for the historical period and the four tier-1 scenarios from ScenarioMIP (O'Neill et al., 2016). The zonal means from the input files are mapped onto the TM5 grid, using a local mass conserving regridding scheme in the vertical direction (and linear interpolation in the meridional direction).

The evolution of methane is constrained by the surface mixing ratio data provided by CMIP6 (Meinshausen et al., 2017; 2020). In the lower troposphere, the mixing ratios of methane are nudged towards zonal mean values calculated from the CMIP6 input data,





using a relaxation time constant of $2.5 \times 10^5$ s or 2.9 days (Bândă et al., 2014). The domain where this nudging is applied extends from the surface to the highest model layer with full-level pressure above 550 hPa for a surface pressure of 984 hPa. Area averaging is applied to coarsen the zonal mean input fields from a meridional resolution of 0.5° to 2° in TM5. In the stratosphere, methane is nudged to a climatology derived from measurements made by the HALOE (Halogen Occultation Experiment) satellite instrument

(Grooß and Russell, 2005) scaled according to the time series of the annual and global mean surface mixing ratio provided by CMIP6. Following the recommendation of Meinshausen et al. (2017), we assume there is a delay of one year between mixing ratios at the surface and in the stratosphere. Because the HALOE measurements were made from October 1991 to August 2002, we assume that the climatology is representative of the 10-year period 1992–2001, which translates to 1991–2000 at the surface. Thus, the scale factor is defined as the ratio of the global mean mixing ratio in a particular year and the average over 1991– 2000.


The ozone and methane mixing ratios from TM5 are input to the SW and LW radiation schemes of IFS. The methane mixing ratios are also used in IFS to determine the production of water vapour by oxidation of methane in the stratosphere. This calculation makes use of the same parameterization as in EC-Earth3 (see Döscher et al., in preparation).

**2.4 Anthropogenic and natural emissions**

This section gives an overview of the emissions of reactive gases and aerosols applied in TM5. The amounts of emissions from anthropogenic activities and open biomass burning are specified using data sets provided by CMIP6 (Feng et al., 2020): historical anthropogenic emissions are taken from the Community Emissions Data System (CEDS; Hoesly et al., 2018), historical fire emissions from the BB4CMIP6 data set (van Marle et al., 2017), and future emissions from the respective scenario data sets

(Gidden et al., 2019). Anthropogenic and biomass burning emissions for the historical period are provided as monthly and annually varying fields. An exception is the anthropogenic emissions of methane prior to 1970, for which monthly emissions are only provided at 10-year intervals. Scenario emissions are provided for 2015 and from 2020 onwards also at 10-year intervals. For these cases, the emissions in intermediate years are calculated by linear interpolation.

Biogenic emissions of non-methane volatile organic compounds (NMVOCs) and carbon monoxide (CO) are prescribed using monthly estimates from the MEGAN-MACC data set (Sindelarova et al., 2014) for the year 2000. Distinct diurnal cycles are applied to the biogenic emissions of isoprene and monoterpenes (Bergman et al., in preparation). Speciated anthropogenic NMVOC emissions are provided for all sources, except for the aircraft sector. Following the recommendations from the CEDS team, the NMVOC emissions from the aircraft sector are split using distinct NMVOC profiles for the contributions from in-flight exhaust

and takeoff and landing. Natural methane emissions and the rate coefficients describing the uptake of methane by soils are prescribed using estimates from Spahni et al. (2000) for the year 2000. The sources of mineral dust and sea salt, the oceanic source of DMS, and the production of nitrogen oxides ($NO_x$) by lightning are calculated online, as described below. All other natural emissions are prescribed as documented in van Noije et al. (2014). These include terrestrial DMS emissions from soils and vegetation, biogenic emissions of $NO_x$ and ammonia ($NH_3$) from soils, oceanic emissions of CO, NMVOCs and $NH_3$, and $SO_2$

fluxes from continuously emitting volcanoes.

The dust source is calculated using the scheme developed by Tegen et al. (2002). It is based on the assumption that a particle can be released from the soil when the surface friction velocity exceeds a certain threshold value, which depends on the size of the particle and the roughness of the surface (Marticorena and Bergametti, 1995). The threshold friction velocity is determined using

a monthly climatology of roughness lengths derived from scatterometer observations from the European Remote Sensing (ERS)





satellite (Prigent et al., 2005; see also Cheng et al., 2008). In grid cells with a substantial fraction of cultivated land, the dust source is enhanced by reducing the threshold friction velocity by up to 27 %, following a similar approach as in Tegen et al. (2004). Currently, the fractional cropland areas assumed in this calculation are based on a data set for 1992 (Ramankutty and Foley, 1999). The vertical flux of dust particles is subsequently calculated as a function of particle size and the 10 m horizontal wind speed

following Tegen et al. (2002). The grassland and shrubland area fractions that enter this calculation are determined from the vegetation fields received from IFS. Snow covered areas are excluded as dust sources; the snow cover is estimated from the snow depth as in Tegen et al. (2002). Soil moisture effects are presently not accounted for.

The vertical flux is calculated for four size bins with radius boundaries at 0.1, 0.3, 0.9, 2.7, and 8.0 μm (actually 8.0 divided by $3^n$

with integer number $n$ running from 4 to 0). The resulting size-resolved flux is subsequently mapped to the accumulation and coarse modes of M7 with mass median radius set to 0.37 and 1.75 μm and geometric standard deviation of 1.59 and 2.0, respectively (Stier et al., 2005). The weights of the distributions are determined such that the respective mass fluxes in the intervals covered by the first bin and by the second to fourth bins are exactly conserved. The mass and number fluxes associated with these two lognormal distributions are put into the insoluble accumulation and coarse modes, respectively. The global dust emission is tuned

by applying a constant correction factor to the threshold friction velocity (Tegen et al., 2004; Cheng et al., 2008). This factor has been set to 0.6 (see Sect. 3).

The source of sea salt is calculated following Gong et al. (2003) with a temperature dependence based on Salter et al. (2015). The flux of the number of sea-spray particles formed over ice-free ocean areas is expressed as a function of the particle radius at 80 %

humidity and the 10 m horizontal wind speed, $U_{10}$, using the parameterization from Gong et al. (2003). In this formulation the ocean whitecap coverage fraction, $W$, is related to $U_{10}$ as $W = 3.84 \times 10^{-6} \times U_{10}^{3.41}$ (Monahan and Muircheartaigh, 1980). This size-resolved flux is approximated by two lognormal distributions with number median dry radius of 0.09 and 0.794 μm and geometric standard deviation of the accumulation and coarse modes of M7, respectively (Vignati et al., 2010a). The weights of the distributions are determined by requiring that the mapping conserves the integrated number fluxes of particles with dry radius in

the ranges 0.05–0.5 and 0.5–5 μm. It is assumed that the emitted sea-spray particles consist of sea salt only. The particle number and sea-salt mass fluxes associated with these two lognormal distributions are put into the soluble accumulation and coarse mode, respectively.

Several studies have indicated that the formation of sea spray depends on the sea water temperature and that this temperature

dependence changes the size distribution of the formed particles (e.g. Mårtensson et al., 2003; Ovadnevaite et al., 2014; Salter et al., 2014). Following the approach of Salter et al. (2015), we describe the temperature effect using distinct multiplication factors for the particle fluxes in the accumulation mode, $f_a$, and coarse mode, $f_c$. These factors are given by:

$f_a(T) = -8.75593 \times 10^{-5} \times T^3 + 5.56771 \times 10^{-3} \times T^2 - 0.11670 \times T + 1.79321$      for $-1 \leq T \leq 15$ °C    (5)

$f_c(T) = 3.75294 \times 10^{-2} \times T + 0.43706$      for $-1 \leq T \leq 30$ °C,    (6)

where $T$ is the sea surface temperature (SST) in degrees Celsius and coefficients have been rounded to five decimals. No temperature dependence is assumed outside the indicated ranges. Equations (5) and (6) are simplified forms of the empirically derived polynomial expressions from Salter et al. (2015) for their modes with number median radii of 0.0475 and 0.75 μm, respectively, which have been scaled to 1.0 at a reference SST value of 15 °C. Figure 1 shows the temperature factors $f_a$ and $f_c$ as functions of the SST.





The DMS flux in ice-free ocean areas can be calculated as the product of the local surface ocean DMS concentration and the gas transfer velocity (e.g. Lana et al., 2011). The ocean concentrations are prescribed according to the monthly climatology from Lana et al. (2011). The gas transfer velocity is parameterized following Wanninkhof (2014). It is proportional to $U_{10}^2$ and depends on the SST through the Schmidt number. The Schmidt number is expressed as a fourth-order polynomial of the SST.


The amount of $NO_x$ produced in lightning discharges is calculated using an improved version of the parameterization described by Huijnen et al. (2010). Compared to earlier model versions (e.g. Huijnen et al., 2010; van Noije et al., 2014), the distribution between cloud-to-ground (CG) and intra-cloud (IC) discharges has been corrected for clouds for which the thickness of the cold sector is less than 5.5 km. Previously, the percentage of CG discharges was set to 0 % for these clouds; this has now been changed to 100

% (Price and Rind, 1994). As a second revision, it is now assumed that IC flashes are as efficient in producing $NO_x$ as CG flashes (Ridley et al., 2005; Ott et al., 2010). The amount of $NO_x$ produced by each flash is still scaled by a constant factor to ensure that the global total production is around 6 Tg N yr$^{-1}$ (see Sect. 3).

As in earlier versions of the model the emissions of black carbon, primary organic aerosols and sulfate are characterized using
lognormal size distributions with the same geometric standard deviation as the corresponding modes in M7. Previously, carbonaceous aerosols were emitted in the Aitken modes of M7 using size distributions with a number median radius of 0.040 μm for open biomass burning and 0.015 μm for all other sources (aan de Brugh et al., 2011). These values were taken from Dentener et al. (2006), without correcting for the fact that the Aitken modes in M7 have a different geometric standard deviation than the distributions recommended in that study. All of the BC emissions were put into the insoluble mode, while 65 % of the emitted
POA mass was assumed to be water soluble irrespective of the source of the emissions (Stier et al., 2005). In the current model version, the carbonaceous emissions are described using an Aitken-mode distribution for the insoluble particles and an accumulation-mode distribution for the soluble particles (see Table 4). Following Stier et al. (2005), the number median radii of these distributions are set to 0.030 and 0.075 μm, respectively. Carbonaceous emissions from solid biofuel burning and open biomass burning are now assumed to have similar characteristics: for both POA and BC it is assumed that 95 % of the mass from
these sources is emitted in the (soluble) accumulation mode (see Sect. 3). BC and POA emissions from other sources are assumed to be insoluble.

Table 4. Distribution of the carbonaceous aerosol emissions from open biomass burning, biofuel burning and other sources over the insoluble Aitken and soluble accumulation modes, as mass percentages.

|  | Insoluble Aitken | Soluble accumulation |
| --- | --- | --- |
| **Black carbon (BC)** |  |  |
| Open biomass burning | 5. | 95. |
| Biofuel burning | 5. | 95. |
| Other sources | 100. | 0. |
| **Primary organic aerosols (POA)** |  |  |
| Open biomass burning | 5. | 95. |
| Biofuel burning | 5. | 95. |





| Other sources | 100. | 0. |
| --- | --- | --- |


Following the recommendation by Dentener et al. (2006), 2.5 % of the emitted $SO_x$ mass is assumed to be emitted in the form of $SO_4$ particles. These particulate emissions are distributed over the soluble Aitken, accumulation and coarse modes, using three lognormal size distributions with number median radii set to 0.030, 0.075 and 0.75 μm, respectively (Stier et al., 2005). The

distribution of the emitted $SO_4$ mass from the various sources and sectors over the three modes has been revised as indicated in Table 5. Previously, 100 % of the emissions from the industrial sector were put into the accumulation mode, and 50 % of the emissions from all other sources and sectors were put into the Aitken mode and the other 50 % in the accumulation mode (aan de Brugh et al., 2011). The new distribution more closely follows the recommendations from Dentener et al. (2006).

Table 5. Distribution of the sulfate emissions from the various sources and sectors over the soluble Aitken, accumulation and coarse modes, as mass percentages.

| | Aitken | Accumulation | Coarse |
| --- | --- | --- | --- |
| Industrial sector | 0. | 50. | 50. |
| Energy sector | 0. | 50. | 50. |
| International shipping | 0. | 50. | 50. |
| Open biomass burning | 0. | 100. | 0. |
| Volcanoes (non-explosive) | 50. | 50. | 0. |
| Other sources and sectors | 100. | 0. | 0. |

All emissions except those from the aircraft sector are provided as two-dimensional fields and are distributed over model layers
using the vertical profiles given in Table A1 of van Noije et al. (2014). Here the emissions from open biomass burning, including those from grassland fires and agricultural waste burning, are distributed according to the profiles defined for forest fires.

## 2.5 Technical and numerical aspects

The atmospheric grid of TM5 is a regular latitude-longitude grid with a resolution of 3° × 2° (longitude × latitude). The base time
step of the model is 1 h, but the time step is dynamically reduced where needed to fulfil the Courant-Friedrichs-Lewy (CFL) stability criterion (Krol et al., 2005; Huijnen et al., 2010). To avoid the need for very short time steps, a reduced grid is applied in the zonal advection routine. In the reduced grid, the number of grid points in the zonal direction gradually decreases when approaching the poles. By merging cells, the number per latitude band is reduced from 120 equatorward of 76° to 40 between 76 and 78°, 8 between 78 and 82°, 4 between 82 and 88°, and 2 between 88 and 90°. The TM5 grid consists of 34 hybrid sigma-
pressure layers in the vertical direction, which have been constructed by merging of layers defined on the IFS grid. Except for the top layer, all layers in TM5 are combinations of two or three adjacent layers in IFS. The top layer corresponds to five layers in IFS and has a full-level pressure of ~ 0.1 hPa.



Details about the OASIS data exchange between IFS and TM5 are given in van Noije et al. (2014). Their Table 1 gives a list of
the atmospheric and surface fields transferred from IFS to TM5. Grid point atmospheric fields are interpolated from the N128
reduced Gaussian grid to the TM5 3° × 2° grid. To make use of the higher resolution in IFS, surface fields are interpolated to 1° ×
1° and applied at this resolution in TM5 to calculate dry deposition velocities and the sources of mineral dust, sea salt and oceanic
DMS. The west-east and south-north components of the 10 m wind are not used by TM5 anymore and have been replaced by the
10 m wind speed; furthermore, the sea surface temperature, needed in the calculation of the production of sea spray and the oceanic
535    DMS flux, has been added as an instantaneous field.

Table 6 lists the fields IFS receives from TM5, and where they are applied. These fields are all three-dimensional and instantaneous.
Mass mixing ratios are passed for all M7 components (including OA, but not POA and SOA separately) as well as nitrate and
MSA. Since nucleation-mode particles can be neglected for cloud activation and LW radiative effects, the associated number and
540    mass mixing ratios are not included in the data transfer. The aerosol optical properties fields are the extinction, single-scattering
albedo and asymmetry factor at the 14 wavelength bands of the RRTMG SW radiation scheme (see Sect. 2.2). We recall that the
model distinguishes between stratospheric and tropospheric aerosols, and that the TM5 aerosol fields are only used by IFS in the
troposphere. The transfer of aerosol fields and the calculation of the optical fields for $RRTMG_{SW}$ are therefore limited to the lowest
23 layers, i.e. the domain extending from the surface to 73.4 hPa. The convection calculations in TM5 are limited to the same
domain.

Table 6. Fields transferred from TM5 to IFS.

| Field | Application in IFS | Domain of application |
| --- | --- | --- |
| Methane mixing ratio | SW and LW radiation scheme | Whole atmosphere |
| Ozone mixing ratio | SW and LW radiation scheme | Whole atmosphere |
| Aerosol number mixing ratio per mode | Cloud activation scheme | Troposphere |
| Aerosol component mass mixing ratios per mode | Cloud activation scheme, LW radiation scheme | Troposphere |
| Aerosol optical properties | SW radiation scheme | Troposphere |

The time interval of the data exchange between IFS and TM5 is 6 h. This is eight times the time step in IFS and six times the base
time step in TM5. To put this into perspective, note that during many years the standalone configuration of TM5 has been driven
by 6-hourly meteorological fields and that the full radiation computations in IFS (cycle 36r4) are done only every 3 h (Morcrette
et al., 2000). It would be possible to increase the exchange frequency to 3 h, but this would lead to a substantial decline of the
computational performance.

Built around the OASIS3-MCT coupler, the synchronization of IFS and TM5 has been overhauled. On the IFS side, it uses the
new coupling interface introduced in EC-Earth3 (Döscher et al., in preparation). The two models still run concurrently but TM5
execution is not delayed by a full coupling interval anymore. Indeed, in the previous implementation, TM5 could start simulating
the next 6-hourly interval only once IFS had reached the end of it (van Noije et al., 2014). The advantage was that TM5 knew the
pressure at the beginning and end of the coupling interval without any lag, and could adjust the horizontal air mass fluxes to close



the air mass balance and ensure mass conservative transport of tracers during that interval (Segers et al., 2002). The new approach removes the 6 h delay, making the execution of TM5 and IFS more synchronous, but introduces a lag (as defined by OASIS3-MCT) of 45 min for the fields received by TM5 (i.e. one IFS time step). Although only the fields at the beginning of the interval are known now, TM5 still applies a mass conservative transport operator. This leads to a surface pressure that slightly diverges from its IFS counterpart but is correctly reset at the start of the next coupling interval. The new design has some important

advantages. It removes the need to run the models sequentially when the feedback from TM5 to IFS is switched on. Moreover, that feedback occurs with a lag of 1 h (one TM5 timestep) instead of 3 h (half a coupling interval) previously. Equally as important, it greatly facilitates coupling additional components to TM5, like a dynamic global vegetation model and/or an ocean biogeochemistry component. This capability was exploited to develop a carbon-cycle configuration of EC-Earth3 (Döscher et al., in preparation).


The large amount of data exchanged with TM5 has always been hindering the model performance. Several steps have been taken to alleviate this issue (see Table 7). First, since TM5 runs on a subset of IFS levels, the reduction operation was moved from TM5 to IFS to decrease the amount of multi-level data transferred to TM5, which halved the execution time. An additional 40 % performance increase was obtained by packing several levels together into one OASIS3-MCT entry, taking advantage of the new

bundle feature of the coupler (Craig et al., 2017). Finally, the parallelization of TM5 with the Message Passing Interface (MPI) library has been revised. The domain decomposition now consists in a geographical partitioning (Williams et al., 2017). This has strongly improved the scalability of the TM5 model, and also enabled all MPI tasks to communicate with OASIS3-MCT, essentially parallelizing the exchange of grid-point fields. In the model version described by van Noije et al. (2014), only one TM5 core was communicating with OASIS3. Although the transfer of spectral fields cannot be distributed in the same fashion, the multi-

core coupling of grid-point fields has improved the performance by 35 %. The last improvement stems from limiting the transfer of aerosol fields to 23 out of 34 levels, which corresponds to the domain where convection is applied in TM5 with a top at about 70 hPa.

Table 7. Computational speed in simulated years per day (SYPD) for different implementations of the data exchange between TM5

and IFS. The benchmark tests were primarily conducted on the high-performance computer of the ECMWF. Performances from additional platforms with the final CMIP6 configuration are also reported.

| Iteration | Computational speed (SYPD) | Platform |
|---|---|---|
| Transfer IFS fields on 91 vertical levels | ~ 0.4 | Cray XC30 (ECMWF) |
| Transfer IFS fields on 34 vertical levels | ~ 0.87 | Cray XC30 (ECMWF) |
| Transfer IFS fields on 34 vertical levels in three bundles | ~ 1.4 | Cray XC30, XC40 (ECMWF) |
| Switch to multi-core coupling (grid-point IFS and TM5 fields) | ~ 2.0 | Cray XC40 (ECMWF) |
| Transfer TM5 aerosol fields on 23 out of 34 levels | 2.3–3.2 | Cray XC40 (ECMWF, SMHI, CSC), Atos Bullx B500 (KNMI), ClusterVision Tetralith (NSC) |



In its latest iteration with load balancing and optimization of bundle sizes, EC-Earth3-AerChem runs at about 3 simulated years per day (SYPD; Balaji et al., 2017) on the most recent platforms. This can be compared to the standard EC-Earth3 model and its
high-resolution configuration, which typically reach 15–20 and 2–4 SYPD (Haarsma et al., 2020), respectively. Clearly, performance-wise, the increased complexity from interactive aerosols and atmospheric chemistry costs about as much as increasing the horizontal resolution of the model (by a factor of 2 and 4 for the atmosphere and ocean components, respectively).

## 3 Tuning and spin-up

As a first step in the tuning process, a small number of parameters in TM5 were optimized in a standalone configuration driven by meteorological and surface fields from the ERA-Interim reanalysis (Dee et al., 2011) using the same horizontal resolution and number of vertical levels as in EC-Earth3-AerChem (see van Noije et al., 2014). Specifically, the correction factor for the threshold friction velocity applied in the calculation of the mineral dust source was set to 0.6, resulting in a global source of $1.12 \times 10^3$ Tg in the year 2010. This is well within the range obtained in other global models (e.g. Huneeus et al., 2011; Gliß et al., 2020), although

for a proper comparison of emitted mass amounts from different models one should account for differences in the representation of the upper end of the size distribution. Moreover, the scale factor applied to the $NO_x$ produced in lightning flashes was determined from the requirement that the total production in 2006 is 6.0 Tg N.

The assumptions about the size distribution and solubility of carbonaceous aerosols emitted from biofuel and open biomass burning
have also been revised as part of the tuning. Initially, 65 % of the organic matter from open biomass burning was assumed to be water soluble, consistent with observations (Mayol-Bracero et al., 2002; Reid et al., 2005). POA emissions from other sources, including solid biofuel burning, as well as freshly emitted BC were assumed to be 100 % insoluble and thus emitted in the Aitken mode. This resulted in too high particle number concentrations in residential regions with substantial biofuel burning. The carbonaceous emissions from solid biofuel burning have therefore been separated from the other emissions in the residential sector
and are treated as emissions from open biomass burning. In an intermediate version of the model, 50 % of the BC mass emitted by biofuel and biomass burning was assumed to be emitted into the (soluble) accumulation mode (see e.g. Kodros et al., 2015). In line with measurements of the size distributions of emissions from open biomass burning (Janhäll et al., 2010) and biofuel burning (Li et al., 2009; Winijkul et al., 2015), this percentage has later been increased to 95 % for both POA and BC. This revision has led to modest reductions in the aerosol optical depth in parts of East Asia and, during boreal winter and spring, in Western Africa and
the tropical Atlantic, improving the comparison with observations in these regions (not shown). An evaluation of aerosol optical properties for the year 2010 simulated with the final parameter settings in TM5 is presented by Gliß et al. (2020). This evaluation includes both the TM5 standalone configuration driven by meteorological and surface fields from the ERA-Interim reanalysis (Dee et al., 2011) and the EC-Earth3-AerChem model in atmosphere-only configuration with sea surface temperatures (SSTs) and sea ice concentrations prescribed as in the Atmospheric Model Intercomparison Project (AMIP) experiment (Döscher et al., in
preparation) and atmospheric temperatures and surface pressures nudged to ERA-Interim fields.

The tuning of the model's climate started from the tuned configuration of EC-Earth3 (Döscher et al., in preparation). As explained in the introduction, the main differences between the two configurations are due to tropospheric aerosols and tropospheric and lower-stratospheric ozone. In EC-Earth3, tropospheric aerosols are described by the MACv2-SP simple plume representation of
anthropogenic aerosol optical properties and cloud effects (Stevens et al., 2017) in combination with a pre-industrial climatology produced by TM5.




EC-Earth3 produces an aerosol effective radiative forcing (ERF) of about -0.8 W m$^{-2}$ over the CMIP6 historical period (1850–2014), as estimated from a set of 30-year atmosphere-only simulations performed as part of the Radiative Forcing Model Intercomparison Project (RFMIP; Pincus et al., 2016). For comparison, using the same IFS parameter settings as in EC-Earth3, the aerosol ERF in EC-Earth3-AerChem was estimated at -1.1 W m$^{-2}$. The final revision of the treatment of carbonaceous emissions from biofuel and biomass burning emissions resulted in a ~ 0.4 W m$^{-2}$ weaker aerosol forcing, bringing the forcing in EC-Earth3-AerChem closer to that in EC-Earth3. This is mainly due to a reduction in the SW cloud forcings, as we have verified using the method proposed by Ghan (2013). (The aerosol ERF estimates for both configurations were obtained from 15-year simulations with AMIP SSTs and sea ice concentrations for the years 2000–2014, as the difference in the net energy imbalance at the top of the atmosphere (TOA) between simulations with emissions for 2000–2014 and 1850, respectively. To isolate the effects of tropospheric aerosols, the mixing ratios of methane and ozone in these simulations were prescribed in IFS as in EC-Earth3.)

In view of these results, our tuning efforts focused on the pre-industrial climate of EC-Earth3-AerChem; no attempt was made to make specific adjustments to improve the model's climate for the present day or the simulated warming over the historical period. When tuning the pre-industrial climate of EC-Earth3-AerChem, a small number of atmospheric tuning parameters in IFS has been re-adjusted, leaving ocean and sea ice parameters in NEMO untouched. The model was initialized from the IFS and NEMO states taken from the EC-Earth3 pre-industrial control simulation (member r1i1p1f1, after 500 years), and a TM5 state representative of pre-industrial conditions. (After 10 years, a small update of the pre-industrial vegetation climatology was introduced. This had only a minor impact on the simulated pre-industrial climate.) Without re-adjusting any tuning parameters in IFS the model started to drift to a new climate state, characterized by higher temperatures especially in the Northern Hemisphere. The increase in zonal mean surface air temperatures varied from less than a few tenths of a degree in the mid-latitudes of the Southern Hemisphere to a few degrees at high latitudes in the Northern Hemisphere, turning the cold biases of EC-Earth3 into warm biases in these regions. Informed by a comparison with the ERA5 reanalysis for the 1980s (Herschbach et al., 2020), corrected for the observed warming since pre-industrial times, we tried to reduce these warm biases by re-adjusting a small set of tuning parameters in IFS. Based on experience gained during the tuning of EC-Earth3 (Döscher et al., in preparation), three parameters were selected affecting both warm and cold regions: ENTRORG, the fractional entrainment (m$^{-1}$) for positively buoyant deep convection divided by the gravitational constant; RSNOWLIN2, which governs the temperature dependence of the autoconversion of ice crystals to snow in large-scale precipitation (Lin et al., 1983); and RLCRIT_UPHYS, the critical cloud droplet radius for the autoconversion of droplets into rain in large-scale precipitation (see Sect. 2.2). Using parameter sensitivities derived from EC-Earth3 atmosphere-only simulations, two combinations of settings were defined corresponding to a target global mean surface cooling of 0.5 and 0.75 °C (see Table 8).

Table 8. Parameter settings for the three IFS parameters that have been re-adjusted for tuning the model's pre-industrial climate. The column labelled 'EC-Earth3-AerChem' contains the values adopted in the CMIP6 configuration of the model, which correspond to a target reduction in the global mean surface temperature of 0.5 °C compared to the configuration with the standard EC-Earth3 settings. The settings indicated in the column labelled 'EC-Earth3-AerChem, cold variant' correspond to a target surface cooling of 0.75 °C.

| Tuning parameter | IFS cycle 36r4 | EC-Earth3 | EC-Earth3-AerChem | EC-Earth3-AerChem, cold variant |
|---|---|---|---|---|
| ENTRORG (s$^2$ m$^{-2}$) | $1.8 \times 10^{-4}$ | $1.7 \times 10^{-4}$ | $1.75 \times 10^{-4}$ | $1.75 \times 10^{-4}$ |



| | | | | |
|---|---|---|---|---|
| RSNOWLIN2 (K$^{-1}$) | 0.025 | 0.035 | 0.030 | 0.029 |
| RLCRIT_UPHYS (m) | Not applied | $8.75 \times 10^{-6}$ | $8.75 \times 10^{-6}$ | $8.84 \times 10^{-6}$ |

Initially, the focus was on the cold variant of the model. A sensitivity simulation for this configuration was started by branching off from the reference simulation with standard EC-Earth3 settings (after about 20 years from the start). As expected, the configuration with adjusted settings produced a colder climate. The Northern Hemisphere was more strongly affected than the Southern Hemisphere: at northern high latitudes, the zonal mean surface air temperature was reduced by more than 2 °C. After another ~ 100 years, a third simulation was started with parameter settings as in the final EC-Earth3-AerChem configuration (see

Table 8). This simulation branched off from the reference simulation. After having completed a few decades, the reference simulation was stopped and the two sensitivity simulations were continued for another ~ 90 years. At that point it was discovered that the correction factor for the dust source was set to 0.7, a value obtained for an intermediate version of EC-Earth3-AerChem, resulting in a reduction of the global source to about 550 Tg yr$^{-1}$ in these simulations. After resetting the factor to the intended value of 0.6, a new set of simulations was launched for the three configurations indicated in Table 8. This increased the dust source

to about $1.1 \times 10^3$ Tg yr$^{-1}$, as verified from the first few years of the simulations. The configuration with a cooling target of 0.5 °C produced satisfactory behavior for the same set of atmosphere and ocean variables considered in the tuning of EC-Earth3 (Döscher et al., in preparation), and was spun up for 300 years. Compared to EC-Earth3, this configuration produced higher, more realistic pre-industrial temperature levels in the Northern Hemisphere, resulting in reduced long-term variability in the global mean surface temperature. While running the CMIP6 historical simulation with the selected parameter settings, another bug was discovered in

the code dealing with the stratospheric aerosols. This bug affected only EC-Earth3-AerChem, and led to spurious warming by absorption of SW radiation in the stratosphere. This resulted in a completely wrong response to large volcanic eruptions. After fixing this bug, the pre-industrial spin-up simulation was continued for another 150 years. The impact of the bug fix on pre-industrial surface climate turned out to be small. This completed the tuning and spin-up of the model, totalling to 770 continuous years for the final configuration (on top of the EC-Earth3 pre-industrial control simulation).

**4 Results**

In this section we present results from some of the core CMIP6 simulations conducted with EC-Earth3-AerChem. Here we only include results from simulations with active ocean and sea ice components. An analysis of the AMIP simulation and AerChemMIP atmosphere-only simulations will be presented elsewhere. The CMIP6 historical simulation is compared against observational datasets, using all four available realizations (see Sect. 4.3). For other experiments, the EC-Earth3-AerChem results presented in

this paper are based on a single realization (r1i1p1f1).

**4.1 Pre-industrial control simulation**

Figure 2 shows the time series of the annual mean global surface air temperature (GSAT) and net radiative flux at the top of the atmosphere (TOA) from the pre-industrial control simulation (piControl). Currently, output from 311 years of piControl is

available. This is sufficient to serve as a reference for our four-member ensemble of coupled historical and future simulations covering 1850–2100, where the different realizations have branching times that lie 20 years apart (see Sect. 4.3). The mean GSAT is 14.05 ± 0.16 °C, where the standard deviation describes the natural interannual variability. The corresponding range for EC-Earth3 is 13.87 ± 0.22 °C, as determined from ensemble member r1i1p1f1, which covers 501 years. Hence, in agreement with the goals set during the tuning phase, EC-Earth3-AerChem is slightly warmer and exhibits lower natural variability than EC-Earth3.

The linear trend in GSAT is -0.075 ± 0.009 °C per century. Most of this negative trend is caused by the relatively large low-





temperature excursion that shows up near the end of the simulated period. The mean TOA flux is $-0.10 \pm 0.25$ W m$^{-2}$. The drift in the TOA flux is statistically insignificant at the p=0.05 level ($2.1 \pm 15.6$ mW m$^{-2}$ per century).

### 4.2 Climate sensitivity

In this section we present estimates of the model's climate sensitivity obtained from the two DECK $CO_2$ perturbation experiments. Figure 3a shows the time series of the annual mean GSAT change in these simulations relative to the unperturbed, pre-industrial control simulation. Here the pre-industrial reference values are given by a linear fit through the corresponding 150-year section of piControl. By definition, the transient climate response (TCR) is calculated as the mean GSAT change in the experiment with atmospheric $CO_2$ concentrations increasing by 1 percent per year (1pctCO2) in a 20-year period centred around the time of $CO_2$

doubling (i.e. simulation years 60–79; e.g. Meehl et al., 2020). This results in a TCR estimate of 2.1 °C, which is slightly lower than the corresponding estimate of 2.3 °C obtained for EC-Earth3 (from 1pctCO2 member r3i1p1f1) and in the middle of the range produced by CMIP6 models. For instance, Meehl et al. (2020) obtained a multi-model mean TCR of 2.0 °C with a standard deviation of 0.4 °C, based on CMIP6 model data available from the Earth System Grid Federation (ESGF) in March 2020.

The model's effective climate sensitivity can be obtained from the experiment with quadrupled $CO_2$ concentrations (abrupt-4×CO2) by linearly regressing the annual mean net TOA flux change versus the annual mean GSAT change (Gregory et al., 2004; Andrews et al., 2012; Meehl et al, 2020; Sherwood et al., 2020), where a consistent definition of change is applied to both variables. Hence, the TOA flux change is corrected for the offset and drift in the corresponding section of the control simulation. Note that neither the TOA flux nor the global temperature shows a statistically significant drift in piControl over this 150-year period. The

effective sensitivity is determined from the regression line as the GSAT change at the point where the net TOA flux change reaches zero, divided by 2.0 to convert to double $CO_2$. We have tested both ordinary least squares (OLS) regression and the Theil-Sen regression method, which is more robust to outliers, and applied these regression methods to the full 150-year period, as well as to restricted periods leaving out the first five (Wyser et al., 2020) or 20 years (e.g. Meehl et al., 2020). All methods produce a sensitivity estimate of 3.9 °C. (OLS regression yields 3.86, 3.92 and 3.92 °C for the 150-, 145- and 130-year periods; the

corresponding estimates from the Theil-Sen regression method are 3.86, 3.89 and 3.85 °C.) As an example, Fig. 3b shows the linear fit obtained with Theil-Sen regression applied to the full 150-year period. The value of 3.9 °C is close to the CMIP6 multi-model mean of $3.7 \pm 1.1$ °C from Meehl et al. (2020), but at the high end of the likely range estimated from multiple lines of evidence in the recent study by Sherwood et al. (2020). Applying the same regression methods to the EC-Earth3 abrupt-4×CO2 experiment (members r3i1p1f1 and r8i1p1f1) results in a sensitivity estimate of around 4.3 °C. The lower estimate for EC-Earth3-

AerChem is consistent with the reduction in TCR, and in better agreement with the assessment by Sherwood et al. (2020).

### 4.3 Evaluation of surface air temperatures in the CMIP6 historical simulation

In this subsection we present surface air temperatures (SATs) from the four available realizations of the CMIP6 historical simulation and evaluate the results against observational datasets. The different ensemble members have been initialized from the

pre-industrial control simulation, using branching times 20 years apart. The first member (r1i1p1f1) has been started from the initial state of piControl, and the second (r2i1p1f1), third (r3i1p1f1) and fourth member (r4i1p1f1) from the state obtained after 20, 40 and 60 years of piControl, respectively.

Figure 4 shows the evolution of the annual mean GSAT in the four integrations, together with the corresponding ensemble median

and mean values and the range bounded by one standard deviation (σ) around the mean. For comparison, the figure also shows the





corresponding time series from the first 165 years of the pre-industrial control simulation. The evolution of the global temperature during the 20th century differs strongly among the four members. Apart from short-term cooling events after large volcanic eruptions, the first, second and fourth members remain relatively close to the pre-industrial mean until about 1950. In contrast, the third member is in a significantly colder state during most of the 20th century. In this period, the spread among the four members

exceeds the range of internal variability displayed by the pre-industrial simulation. The ensemble median is higher than the mean for most years in the period 1920 to 1960. In the earlier and later periods, the differences between the median and mean are small. The mean GSAT in the final 10 years of the historical simulation (2005–2014) is 14.76 ± 0.19 °C, where the standard deviation indicates the ensemble spread. This is 0.71 °C above the pre-industrial mean.

In Fig. 5 the median, mean and 1-σ range of annual temperature anomalies are compared against the GISS Surface Temperature Analysis (GISTEMP) version 4 (Lenssen et al., 2019; GISTEMP Team, 2020) and version 2.0 of the temperature reconstruction by Cowtan and Way (2014; 2020). Both data sets combine SAT anomalies over land and sea ice with SST anomalies over open sea. Whereas surface air and water temperatures may be very different, their anomalies are very similar over open sea. We can therefore directly compare the SAT anomalies simulated by the model with the reconstructed anomalies. A more robust comparison

would use a blend of air and water temperatures also from the model (Cowtan et al., 2015), but such an analysis is beyond the scope of this paper. The anomalies shown in the figure are calculated with respect to the period 1850–1900 or, for GISTEMP, 1880–1900. The simulated mean GSAT in this period is 14.15 ± 0.08 °C, almost 0.1 °C higher than the average over the pre-industrial control simulation.

Figure 5a shows that the ensemble of four realizations tends to underestimate the observed global temperature anomalies from the end of the 19th century onwards. The upper end of the 1-σ range follows the observational time series reasonably well until the middle of the 20th century. In contrast to the observations, the warmer members show a substantial global cooling during the 1950s and 1960s (see Fig. 4), partly caused by the eruption of Mount Agung in 1963. As a result, all members produce negative anomalies from the 1960s to the end of the 1980s, while the observed anomalies remain positive during these years. In the final decades of

the historical period, the simulations tend to overestimate the observed warming trend. The mean GSAT anomaly for the years 2005–2014 is 0.62 ± 0.23 °C. For this period, the GISTEMP and Cowtan and Way time series give a global warming of 0.87 and 0.89 °C, respectively. As can be seen in Fig. 5b and c, the spread among ensemble members as well as the discrepancies with the observed time series are almost entirely caused by variability in Northern Hemisphere (NH) temperatures. The simulated temperature anomalies for the Southern Hemisphere (SH) agree rather well with the GISTEMP time series. The ensemble mean

for the period 2005–2014 is 0.69 ± 0.04 °C, compared to 0.65 °C in GISTEMP.

We have verified that the climate states characterized by anomalously low temperatures in the NH are associated with periods of reduced convective mixing in the Labrador Sea, as diagnosed from the local mixed-layer depth (Griffies et al., 2016; not shown). The convective activity is also strongly reduced during the last ~30 years of the pre-industrial control simulation. This suggests

that the model suffers from the same instability mechanism active in other EC-Earth3 configurations displaying spurious interdecadal variability in pre-industrial and historical simulations (Döscher et al., in preparation; Parsons et al., 2020). As discussed in Sect. 5, we believe this instability is related to the use of the NEMO3.6 ocean model and the relatively coarse ORCA1 grid (Koenigk et al., 2020).





Next, we evaluate the ensemble mean SAT climatology for the last 20 years of the historical simulation (1995–2014) using the ERA5 reanalysis from the ECMWF (Hersbach et al., 2020; Copernicus Climate Change Service, 2017) as the observational reference. In Fig. 6 we compare the climatological SAT distributions for all seasons, and for boreal winter (December, January and February) and summer (June, July and August) separately. The ensemble mean GSAT bias with respect to ERA5 for this period is $0.22 \pm 0.18$ °C. The mean SAT bias is $1.29 \pm 0.05$ °C in the Southern Hemisphere and $-0.86 \pm 0.35$ °C in the Northern

Hemisphere. The strongest biases are observed over the Southern Ocean and Antarctica, where temperatures are overestimated throughout the year. Warm biases also exist in the subtropical marine stratocumulus regions in the eastern South Atlantic and the eastern North and South Pacific. These biases are common to many climate models and have been attributed to biases in SW cloud radiative effects (Bodas-Salcedo et al., 2014; 2016, Calisto et al., 2014; Forbes and Ahlgrimm, 2014; Hogan et al., 2017). Temperatures are also overestimated over the Gulf of Alaska and the Bering Sea. Moreover, a warm bias is found in the Kazakhstan

region, mostly during winter. The model underestimates the seasonal cycle in northeastern Siberia, resulting in a warm bias during winter and a cold bias during summer. Strong cold biases exist during winter over the Arctic Ocean, the Labrador Sea and the North Atlantic, and also in the Middle East, northern Africa, the United States. As the NH is warming faster than observed, most of these cold biases tend to be reduced when moving towards the end of the historical period. For instance, restricting the comparison to the final 10 years of the simulation (2005–2014) reduces the mean bias in the NH to $-0.68 \pm 0.35$ °C.


As the historical simulation ensemble underestimates the warming of the NH (Fig. 5b), cold biases in the NH tend to be smaller in the model's pre-industrial climate. Based on GISTEMP and ERA5, we can estimate that the pre-industrial control simulation underestimates the mean NH SAT of the late 19th century (strictly 1880–1900) by $-0.2$ °C. On the other hand, the good agreement between the simulated and observed SH warming implies that the mean SH SAT bias in the model's pre-industrial climate is

similar as for the present day. Our best estimate of this bias based on GISTEMP and ERA5 is $1.2$ °C.

### 4.4 Historical and future perturbation experiments

Figure 7 shows the evolution of the global surface air temperature and net TOA flux from 1850 to 2100 for various CMIP6 and AerChemMIP experiments. Shown are the annual mean temperature and flux perturbations from a single realization of these

experiments (member r1i1p1f1) relative to the pre-industrial means given in Sect. 4.1. The annual global mean tropospheric aerosol optical depth (AOD) at 550 nm for these simulations is presented in Fig. 8. For the historical period, the CMIP6 historical simulation is compared with the AerChemMIP hist-piNTCF experiment, in which the emissions of near-term climate forcers (NTCFs) are kept at pre-industrial levels. NTCFs include aerosols and precursors of aerosols and ozone. Although being an ozone precursor, the AerChemMIP definition of NTCFs does not include methane. The mean tropospheric AOD at 550 nm is 0.094 in

the pre-industrial control simulation and increases to 0.135 at the end of the historical simulation (2005–2014). The hist-piNTCF experiment yields significantly higher global temperatures than the standard historical simulation from the early 20th century onwards. The two simulations diverge most rapidly during the 1950s and 1960s, due to a cooling in the historical simulation. The maximum 10-year rolling mean difference between the two runs is 1.1 °C. However, as we have seen in the previous section, the model produces large interdecadal variability in the historical period, and the cooling in the 1950s and 1960s is likely a reflection

of this. To be able to separate the effects of the NTCF forcings, ensemble simulations are required, which are in production. The time series of the TOA fluxes show large interannual variability with strong downward excursions after major volcanic eruptions, reaching 0.7 and 1.0 W m$^{-2}$ at the end of the historical period (2005–2014) in the historical and hist-piNTCF simulations, respectively.





Projections are given for three scenarios, which only differ with respect to assumptions regarding NTCF emissions and methane concentrations. The first scenario experiment (ssp370) follows the standard SSP3-7.0 shared socio-economic pathway from ScenarioMIP (O'Neill et al., 2016), which has relatively high emissions of NTCFs (Gidden et al., 2019). For this scenario EC-Earth3-AerChem produces a global warming of 4.9 °C towards the end of the century (2091–2100). The second is the AerChemMIP experiment ssp370-lowNTCF, which follows the same socio-economic pathway but with reduced emissions of

NTCFs (Gidden et al., 2019). This scenario produces enhanced warming, reaching 5.4 °C in the final decade. Finally, in ssp370-lowNTCFCH4 it is assumed that both NTCF emissions and $CH_4$ concentrations will be reduced compared to ssp370. This scenario produces a significantly lower warming of 4.4 °C. Thus, for these scenarios, reductions in $CH_4$ concentrations more than offset the enhanced warming due to reductions in NTCF emissions. A more detailed multi-model analysis for these scenario experiments is presented by Allen et al. (2020).

**5 Discussion and conclusions**

This paper documents the global climate model EC-Earth3-AerChem. EC-Earth3-AerChem is a configuration of the EC-Earth3 family of models with interactive tropospheric aerosols and atmospheric chemistry. It uses a coupling between IFS and TM5 to simulate aerosol-radiation and aerosol-cloud interactions as well as chemistry-climate interactions and radiative effects of ozone and methane. We have described the model with a focus on the specific features of EC-Earth3-AerChem compared to the other

EC-Earth3 configurations (Döscher et al., in preparation), and on the updates and improvements introduced in TM5 and the TM5-IFS coupled system since the publication of EC-Earth 2.4 (van Noije et al., 2014).

The model's pre-industrial climate was analysed from the available 311 years of the piControl simulation. The net TOA energy imbalance for these years is only $-0.10 \pm 0.25$ W m$^{-2}$ and shows no statistically significant drift. The global surface air temperature

(GSAT) is on average $14.05 \pm 0.16$ °C and shows a small linear trend of $-0.075 \pm 0.009$ °C per century. Most of this negative trend is due to a low-temperature excursion that appears near the end of the simulated period.

The model's effective equilibrium climate sensitivity (ECS) was robustly estimated at 3.9 °C, using 130 to 150 years of an abrupt-4×CO2 experiment. This is close to the CMIP6 multi-model mean of $3.7 \pm 1.1$ °C presented by Meehl et al. (2020), but at the high

end of the likely range estimated in the recent study by Sherwood et al. (2020). Similarly, the model's transient climate response (TCR) was calculated from a 1pctCO2 experiment, resulting in a value of 2.1 °C. Again, this is close to the CMIP6 multi-model mean of $2.0 \pm 0.4$ °C from Meehl et al. (2020).

A four-member ensemble of the CMIP6 historical simulation shows large interdecadal variability in Northern Hemisphere and

global temperatures during the 20th century, resulting in a large spread among the different members. The ensemble mean GSAT in the last decade of the simulation (2005–2014) is $14.76 \pm 0.19$ °C, which is 0.71 °C above the pre-industrial mean. Here the standard deviation indicates the spread in the ensemble.

The evolution of annual mean surface air temperature (SAT) anomalies in the historical ensemble has been compared with

observational time series from Cowtan and Way (version 2.0) and GISTEMP (version 4). The model underestimates the warming over the historical period. The ensemble mean temperature change between the second half of the nineteenth century (1850–1900) and the period 2005–2014 is $0.62 \pm 0.23$ °C in the simulations compared to 0.87 and 0.89 °C in the GISTEMP and Cowtan and





Way data sets, respectively. The observed warming of the Southern Hemisphere is well reproduced by the model. For the SH, the ensemble mean SAT anomaly for the period 2005–2014 is 0.69 ± 0.04 °C, compared to 0.65 °C in GISTEMP.


The mechanism underlying the spurious interdecadal variability in our simulations is at least partly related to the intermittent nature of the convection in the Labrador Sea, where a reduction or shutdown of the convection during extended periods of time causes anomalously low temperatures in the Northern Hemisphere. Other EC-Earth3 configurations also display spurious interdecadal variability in pre-industrial and historical simulations (Döscher et al., in preparation). We believe this instability is related to the

use of NEMO3.6 and the relatively coarse resolution of the ORCA1 grid (Koenigk et al., 2020). Parsons et al. (2020) examined interdecadal GSAT variability in pre-industrial control simulations from 39 CMIP6 models. The six models showing the largest variability are EC-Earth3, BCC-CSM2-MR, CNRM-ESM2-1, EC-Earth3-Veg, CNRM-CM6-1 and IPSL-CM6A-LR. Five of these use NEMO3.6 on the ORCA1 grid (EC-Earth3, EC-Earth3-Veg) or extended ORCA1 (eORCA1) grid (CNRM-ESM2-1, CNRM-CM6-1, IPSL-CM6A-LR). For future research, it would be interesting to consider increasing the resolution of the ocean model in

EC-Earth3-AerChem.

The cooling of the Northern Hemisphere simulated in the 1950s and 1960s may also be caused or enhanced by aerosol effects. To what extent this is the case needs further investigation. Simulations that provide more information on the role of aerosols and their effective radiative forcing contributions are in production.


The ensemble mean SAT climatology for the years 1995–2014 has been evaluated against ECMWF's ERA5 reanalysis. The GSAT bias for this period is 0.22 ± 0.18 °C. The mean SAT bias is 1.29 ± 0.05 °C in the SH and -0.86 ± 0.35 °C in the NH. As the ensemble overestimates the NH warming trends during this period, cold biases in the NH tend to be reduced towards the end of the simulation. For instance, for the last 10 years of the simulation (2005–2014), the mean NH cold bias is reduced to 0.68 ± 0.35

°C.

Cold biases in the NH also tend to be smaller in the model's pre-industrial climate. The pre-industrial control simulation underestimates the mean NH SAT of the late 19th century by approximately -0.2 °C. The good agreement between the simulated and observed warming of the SH over the historical period implies that the mean SH SAT bias in the model's pre-industrial climate

is similar to the present day. Our best estimate of this bias is 1.2 °C.

Over the Southern Ocean and Antarctica strong warm biases are found in all seasons. Temperatures are also overestimated in subtropical marine stratocumulus regions. These biases are common to many climate models, including all EC-Earth3 configurations, and have been attributed to biases in SW cloud radiative effects. Modifications in the cloud scheme and the

representation of supercooled liquid water made in more recent versions of IFS, including cycle 45r1 (Forbes and Ahlgrimm, 2014; Forbes et al., 2016), together with the introduction of the new ecRad radiation scheme in cycle 43r3 (Hogan et al., 2017) have been shown to substantially reduce these biases.

To illustrate the applicability of the model, time series of the GSAT and net TOA flux change relative to their pre-industrial levels

have been presented for a number of historical and scenario perturbation experiments from AerChemMIP. For the historical period, the perturbation experiment is one with emissions of near-term climate forcers (NTCFs) fixed to pre-industrial levels. This experiment (hist-piNTCF) produces significantly higher temperatures during the 20th century than the standard historical



simulation. For the future period, the standard shared socio-economic pathway SSP3-7.0 from ScenarioMIP has been compared with a scenario with lower NTCF emissions (SSP3-7.0-lowNTCF) and one with both lower NTCF emissions and lower methane concentrations (SSP3-7.0-lowNTCFCH4). For SSP3-7.0, the model projects a global warming at the end of the century (2091–2100) of about 4.9 °C above the pre-industrial level. For the scenario with reduced NTCFs the warming is increased by about 0.5 °C, while for the scenario with both NTCFs and methane reduced the warming is reduced by about 0.5 °C. Note that these estimates are based on a single realization of each experiment. Ensembles required to reduce the impact of internal variability are in production.

The representation of aerosols and chemistry has been updated in numerous respects compared to the model version presented by van Noije et al. (2014). The overall result of these changes is a much improved description of in particular aerosol concentrations and optical properties (Bergman et al., in preparation; Gliß et al., 2020; Checa-Garcia et al., 2020). One highlight worth mentioning is that EC-Earth3-AerChem produces substantially higher and more realistic aerosol optical depths (AOD). The mean tropospheric AOD at 550 nm increases from 0.094 in the pre-industrial control simulation to 0.135 in the last decade of the historical simulation (2005–2014). A detailed evaluation of the aerosol simulation is beyond the scope of this paper.

The increased complexity from interactive aerosols and atmospheric chemistry comes at the expense of computational performance. The single-core OASIS transfer of spectral fields from IFS to TM5 has been identified as a major bottleneck limiting the scalability of the model. It is expected that a substantial speed-up can be achieved by converting the spectral fields to grid-point fields at the IFS side, enabling domain decomposition and multi-core transfer to TM5. This is planned as part of the model's near-term development.

Meanwhile, a number of developments aiming to improve the representation of aerosol and chemical processes are underway. These include the replacement of EQSAM by ISORROPIA II (Fountoukis and Nenes, 2007) or a light version thereof, which is expected to improve the calculation of aerosol water and acidity, as well as the inclusion of coarse-mode nitrate. The MOGUNTIA gas-phase chemistry mechanism has recently been introduced as a more explicit alternative to the current scheme based on CB05 (Myriokefalitakis et al., 2020). A parameterization of marine organic aerosol emissions is also available. Other developments aim to improve the representation of the mineralogical composition and size distribution of dust (Perlwitz et al., 2015a, 2015b; Pérez García-Pando et al., 2016; Adebiyi and Kok, 2020) along with the associated effects upon radiation and clouds. Furthermore, it is envisaged that the cloud activation scheme from Abdul-Razzak and Ghan (2000) will be replaced by the more accurate parameterization developed by Morales Betancourt and Nenes (2014), using a more sophisticated turbulence dependent calculation of the updraft velocity.



**Code availability**

Access to the model code is restricted to institutes that have signed a memorandum of understanding with the EC-Earth community and a software license agreement with the ECMWF. Confidential access to the code can be granted for editors and reviewers.

**Data availability**

The CMOR compliant model output files from which the results presented in this article were calculated will be published on the Earth System Grid Federation (ESGF). Specific fields can be made available by the authors upon request.

**Author contribution**

The core team responsible for the development of EC-Earth3-AerChem consisted of TvN, TB and PLS. Other co-authors contributed to the development of specific aspects of the model or parts of the code that are shared with EC-Earth3. PLS, JPK, DOD, and MGA produced the four members of the historical simulation. All other simulations presented in this paper were set up and carried out by PLS. TvN wrote the paper with input from co-authors.

**Competing interests**

The authors declare that they have no conflict of interest.

**Acknowledgements**

TvN, TB and PLS would like to acknowledge funding from the European Union's Horizon 2020 research and innovation
programme under grant agreement No 641816 (CRESCENDO). The development of EC-Earth3 and EC-Earth3-AerChem has benefitted from services provided by the IS-ENES3 project, which received funding from the European Union's Horizon 2020 research and innovation programme under grant agreement No 824084. JPK and RM wish to acknowledge CSC – IT Center for Science, Finland, for software support and computational resources. MGA and CPGP acknowledge funding by the European Research Council (grant agreement No. 773051, FRAGMENT), the AXA Research Fund, the Spanish Ministry of Science,
Innovation and Universities (RYC-2015-18690 and CGL2017-88911-R), and PRACE and RES for awarding access to MareNostrum at Barcelona Supercomputing Center. RS acknowledges financial support from the strategic research area MERGE (Modelling the Regional and Global Earth system).





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





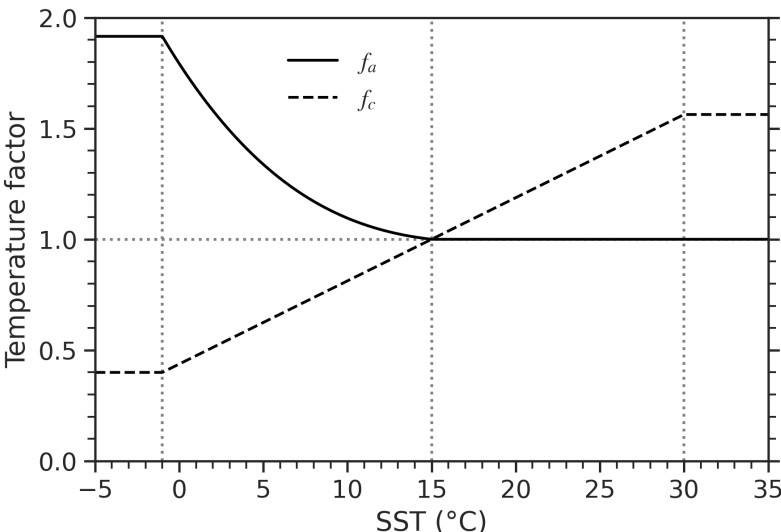

**Figure 1: Temperature factors $f_a$ and $f_c$ for the formation of sea-spray particles in the accumulation and coarse mode, respectively.**



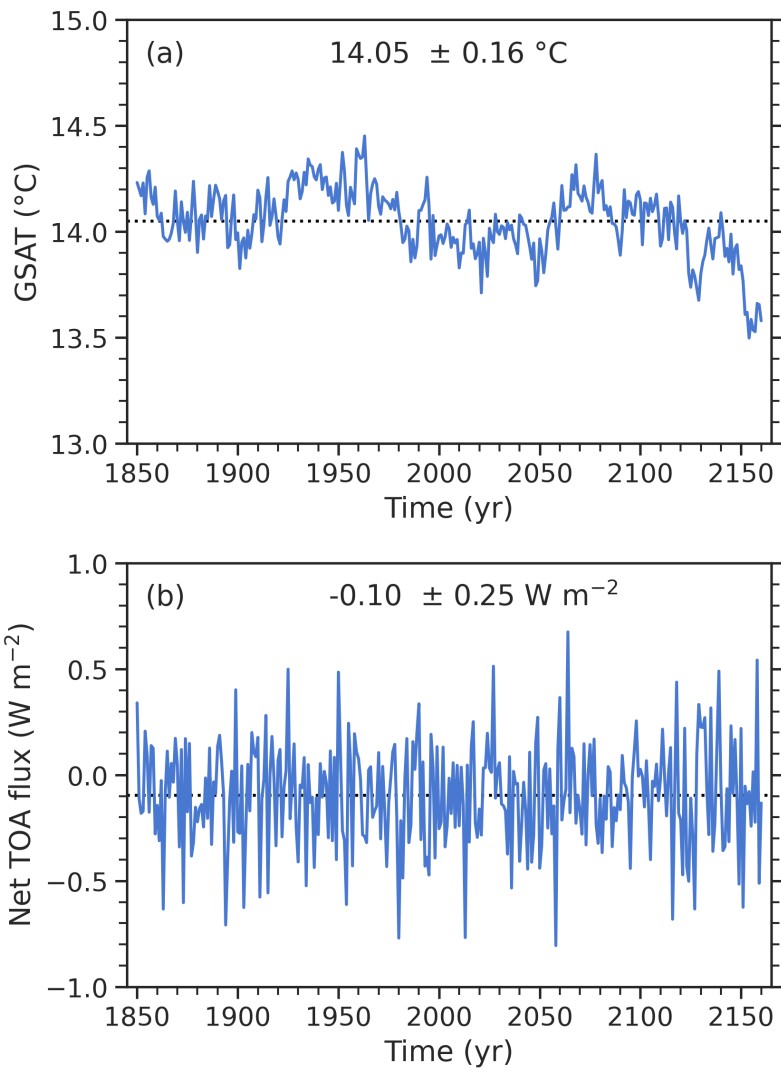

**Figure 2: Time series of (a) the global surface air temperature (GSAT) and (b) the net radiative flux at the top of the atmosphere (TOA) in the pre-industrial control simulation (piControl). Shown are the annual mean values, and the corresponding means and standard deviations.**

1485



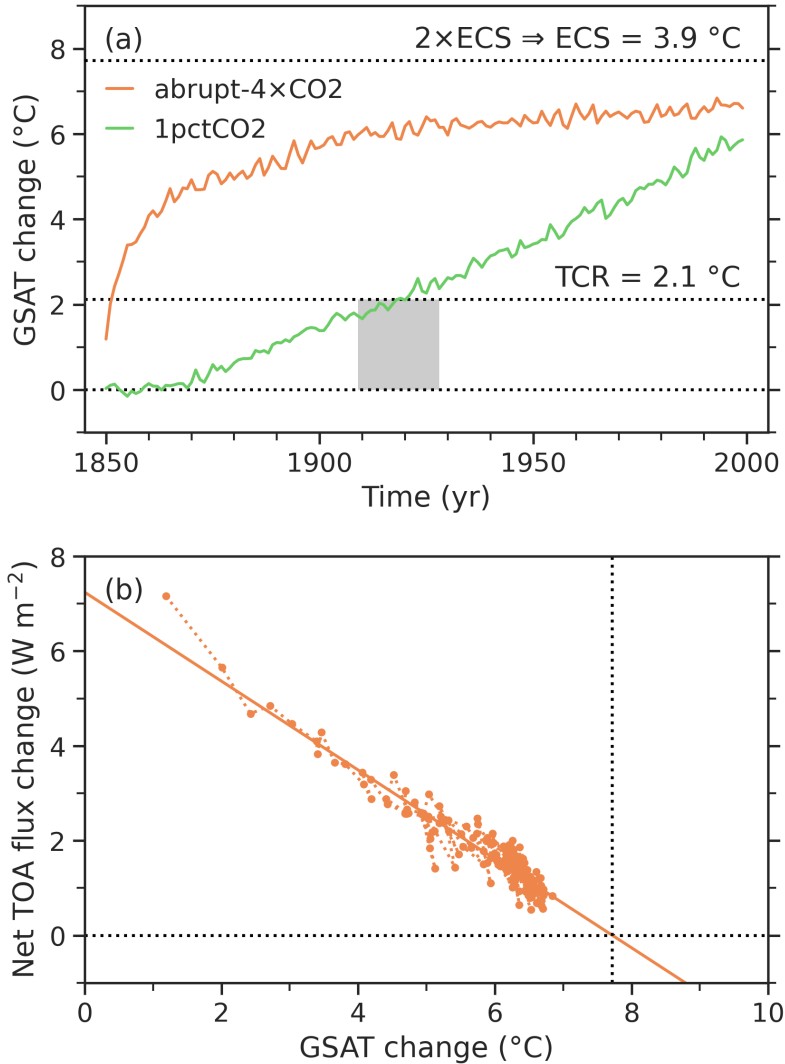

**Figure 3: (a) GSAT change in the simulation with abrupt quadrupling of $CO_2$ concentrations (abrupt-4×CO2) and the simulation with a 1 percent $CO_2$ concentration increase per year (1pctCO2); (b) Net radiative flux change at the top of the atmosphere versus GSAT change in the abrupt-4×CO2 experiment, together with a linear fit obtained by Theil-Sen regression. The temperature and flux changes shown in this figure are the annual mean deviations from the linear trend lines through the corresponding 150 years of the pre-industrial control simulation.**

1490



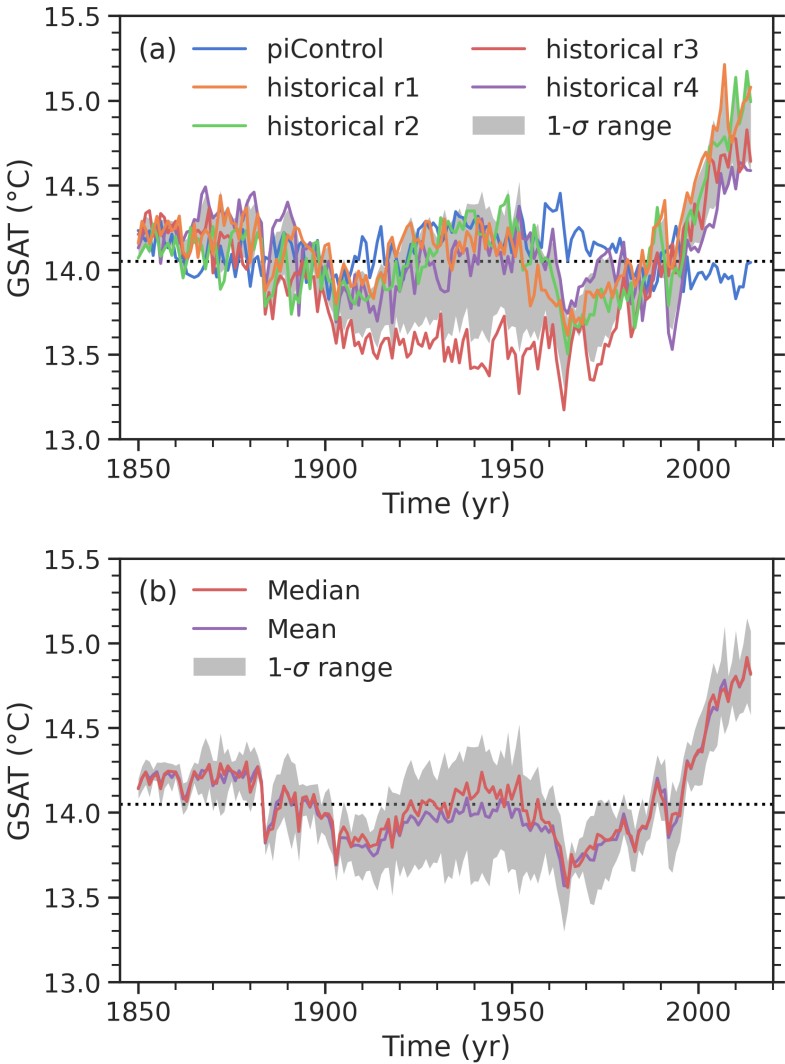

**Figure 4: Time series of the annual and global mean surface air temperature for the four realizations of the CMIP6 historical simulation (a), and the corresponding ensemble median and mean values (b). In both panels, the grey area indicates the range bounded by one standard deviation around the ensemble mean, and the dotted line indicates the pre-industrial mean of 14.05 °C. For comparison, panel (a) also shows the corresponding time series for the first 165 years of the pre-industrial control simulation.**

1495

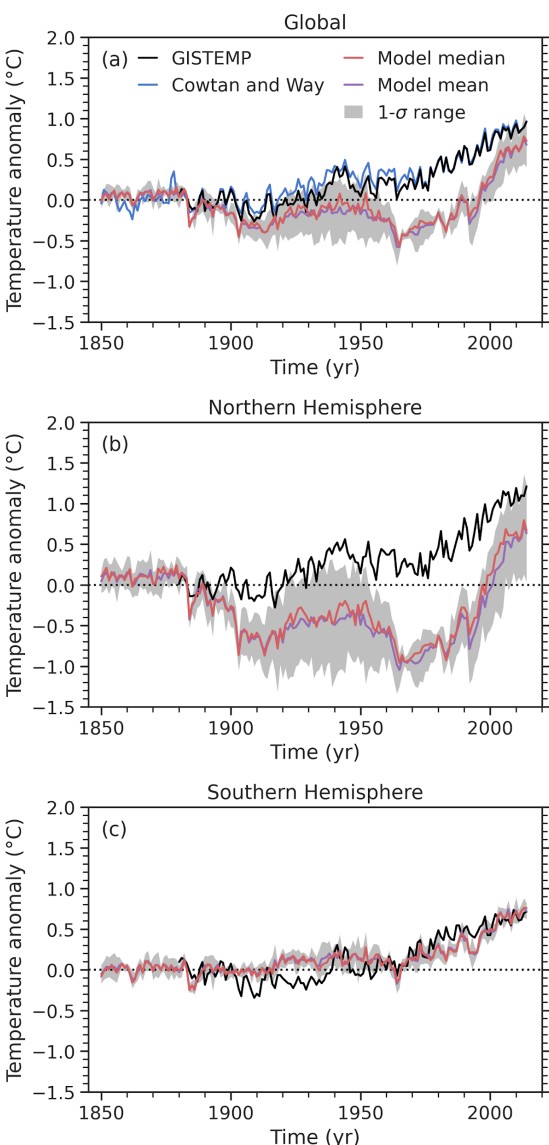

**Figure 5: The median, mean and one-standard deviation range of the annual mean surface air temperature (SAT) anomalies from the four realizations of the CMIP6 historical simulation compared to reconstructed surface temperature anomalies from GISTEMP version 4 and HadCRUT4 infilled by kriging (Cowtan and Way version 2.0). The anomalies are defined with respect to the period 1850–1900 (1880–1900 for GISTEMP). Panels (a), (b) and (c) show the global, Northern Hemisphere, and Southern Hemisphere means, respectively.**

1500

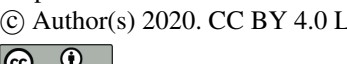



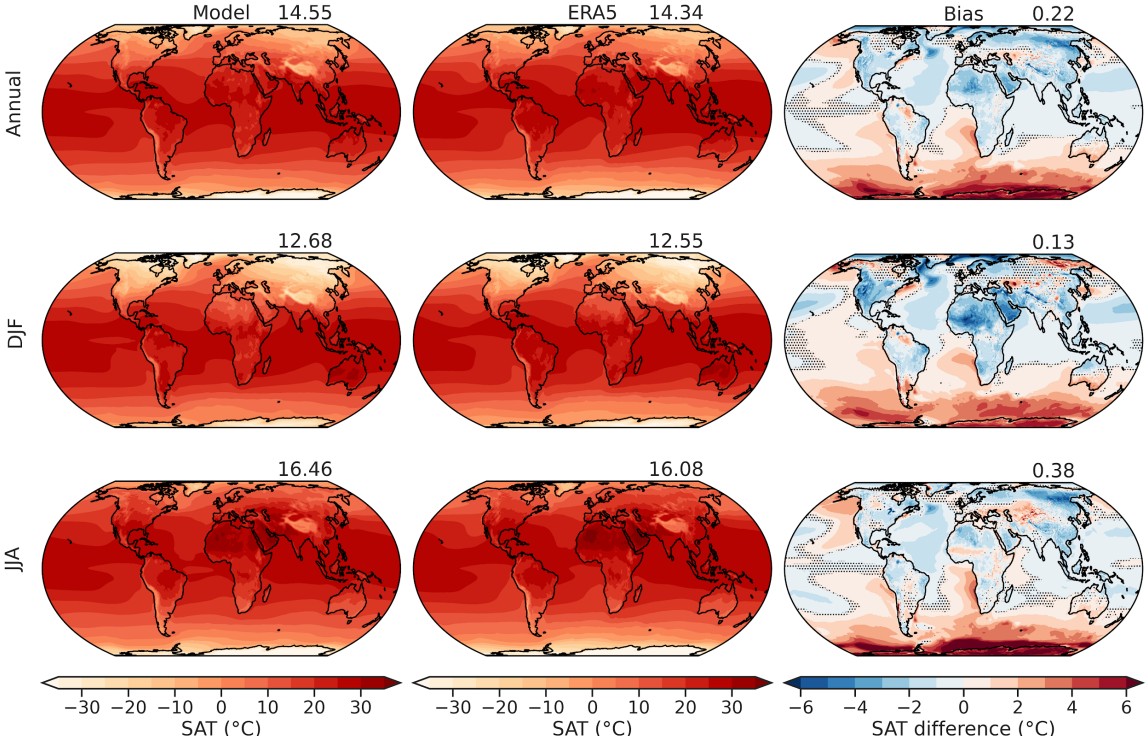

**Figure 6: Surface air temperature (SAT) climatology for the years 1995–2014 from the four-member ensemble of the CMIP6 historical simulation and the ERA5 reanalysis (version 2), and the corresponding differences. The top row shows the full multi-annual means, the middle and bottom rows the means for boreal winter (December, January, February; DJF) and summer (June, July, August; JJA), respectively. The number given at the top right of each panel is the global mean value. The stippled areas in the panels on the right indicate the regions where the differences are not significant at the 5 % level.**


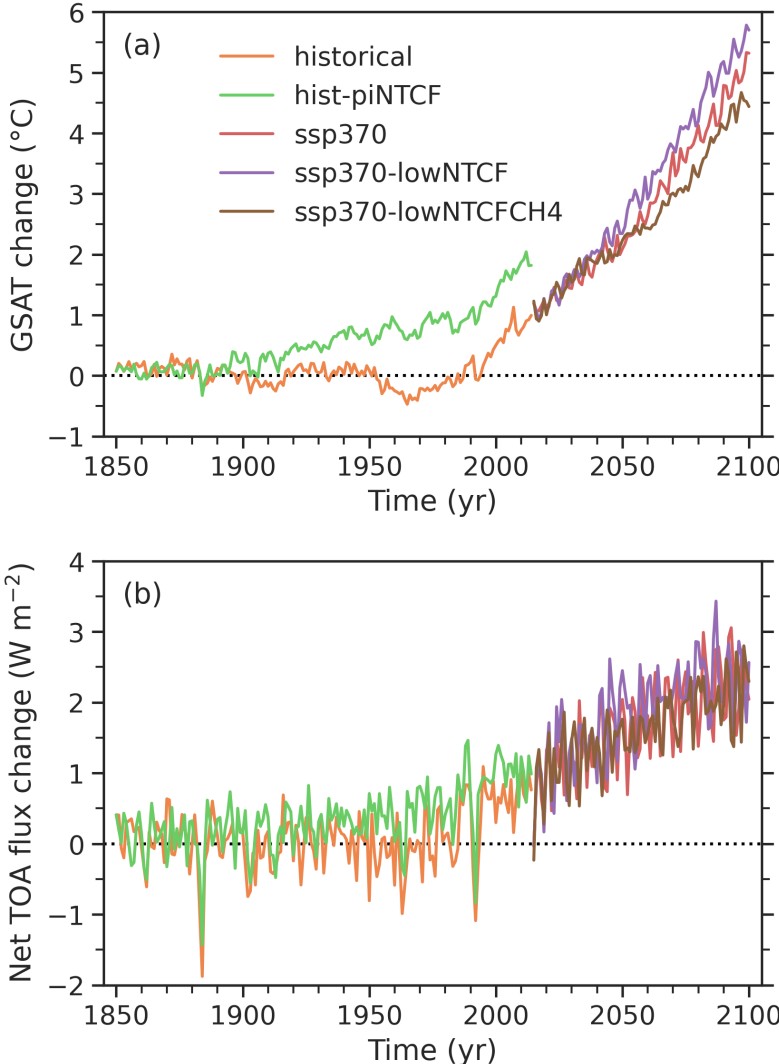

**Figure 7: Change in (a) the global surface air temperature (GSAT) and (b) the net TOA flux in the CMIP6 historical simulation, the AerChemMIP historical simulation with emissions of near-term climate forcers (NTCFs) fixed to pre-industrial levels (hist-piNTCF), and future simulations for the standard SSP3-7.0 scenario, a corresponding scenario with low NTCF emissions (SSP3-7.0-lowNTCF), and a corresponding scenario with both low NTCF emissions and low CH4 concentrations (SSP3-7.0-lowNTCFCH4). The temperature and flux changes shown in this figure are the annual mean deviations from the pre-industrial means given in Fig. 2. Only the first ensemble member (r1i1p1f1) of each experiment is presented.**





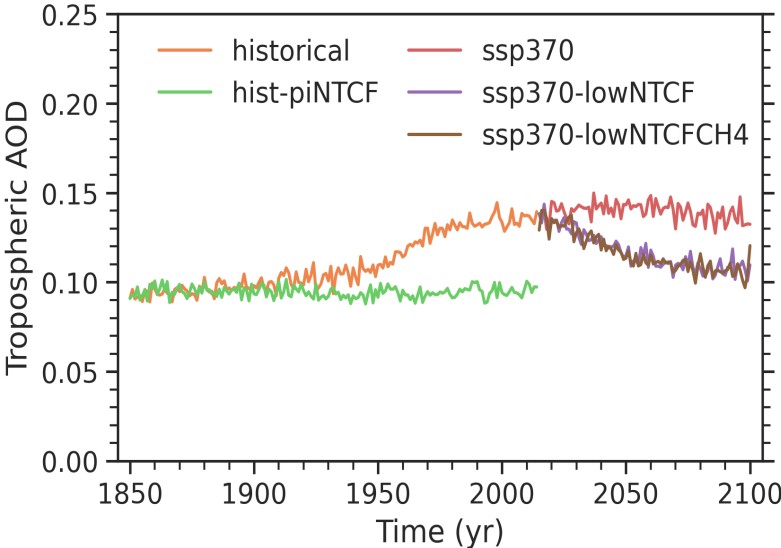

**Figure 8: Annual global mean tropospheric aerosols optical depth (AOD) at 550 nm in the simulations presented in Fig. 7.**