# Peer review of "EC-Earth3-AerChem, a global climate model with interactive aerosols"

_Geoscientific Model Development, 2020_

## Referee Comment (RC1) · Anonymous Referee #1 · 15 Feb 2021

**1   General**

The paper "EC-Earth3-AerChem, a global climate model with interactive aerosols and atmospheric chemistry participating in CMIP6" presents the configuration of the EC-Earth3 climate model used for the AerChemMIP CMIP6 simulations. This configuration adds interactive tropospheric aerosols and tropospheric chemistry to the standard configuration. A thorough description of the aerosol scheme is given followed by an analysis of a few diagnostics, mainly the surface air temperature, from some CMIP6 simulations. As a general comment, I would like to emphasize that the whole paper is very clearly written, and the reader goes through numerous details easily, in a very

fluent way. This a far from being an obvious exercise, congratulations to the authors for that.

This being said, I would like to distinguish two parts in my review.

Firstly, my comments concern the descriptions of the model, and more specifically those of the aerosol characteristics and their interactions with the rest of the climate model. The CMIP6 version builds upon previous versions, in particular that described in van Noije et al. 2014, and the new features are presented here with very precise and specialized details. This is of great interest to those analyzing outputs of this scheme and evenmore to the whole aerosol modelling community. The number of new features in the CMIP6 version is truly impressive. I only have a few suggestions and or questions that I list below in the following section.

Secondly, I would like to indicate that the results shown are not so convincing and would benefit from additional analyses. Results concerning global surface air temperatures in the CMIP6 historical simulation are shown from 4 realisations, three of which show quite large interdecadal variability, one shows quite a substantial cooling during most of the 20th century, and none shows a warming trend comparable to that of observations at the end of the period.

The authors note this "spurious interdecadal variability" and indicate that it has been seen in other EC-Earth3 configurations described in Doscher et al. GMDD, 2021. They point that the instability is related to the use of NEMO3.6 ocean model and ORCA1 grid. This explanation do not seem fully appropriate as for instance the CMIP6 CNRM climate models (CNRM-CM6-1 and CNRM-ESM2-1), that share the same ocean model and a comparable ocean grid, show evolutions of the global surface temperature consistent with observations throughout the 20th century.

Later on, the authors indicate that "The cooling of the Northern Hemisphere simulated in the 1950s and 1960s may also be caused or enhanced by aerosol effects... Simulations that provide more information on the role of aerosols and their effective radiative

forcing contributions are in production".

I would strongly urge the authors to present further analyses on these issues, especially concerning the role of the aerosols. Furthermore, such a large spread in the historical simulations raises doubts on evolutions of the other simulations shown. Therefore, analysis of additional members of both the historical simulation, but also the hist-piNTCF simulation, and the scenario simulations ssp370 and ssp370-lowNTCF would be of benefit to the paper. The authors indicate that such additional members are under production.

Finally, the impact of the complex aerosol scheme should be better highlighted. Either this impact is important or not, positive or not, in the model behaviour, but this should be made clear in the paper.

**2 Comments/questions**

- in addition to information of Table 1, it would be of value to add a table with extinction, single- scattering albedo and asymmetry factors, at 550 nm and 80% humidity for instance

- it is not so clear to me how you deal with stratospheric ozone:

  line 61 you indicate that there is a difference between EC-Earth3 and EC-Earth3-AerChem in "lower-stratospheric ozone": why is that?

  and line 383: how do you deal with differences in the tropopause between your model and the CMIP6 ozone data?

- line 215: please indicate how the wet deposition of aerosol is performed, from convective precipitation and from stratiform precipitation, considering 3D precipitation fields from IFS or...?

- line 338: could you detail why you use fixed values of CDNC here?

- line 423: it would be interesting to have a Table summarizing emitted quantities

- line 502: I could not find in Dentener et al. 2005 recommendation for emission of SO4. Please provide further details

- line 544: you use a fixed number of layers, independently of the tropopause height?

- line 552: have you done any testing for 3h exchange of fields?

- line 608: please indicate names of such regions you observed such a discrepancy in particle number concentration

- line 620: it is more common to nudge winds. Do you have a reference for such a configuration?

- line 780: a usual climatology covers a 30-year period. Why did you restrict it to a 20-year period?

- line 784: biases over Antarctica seem to be larger than 6K, please specify the maximun biaises

- Table 6: could you add fields transferred from IFS to TM5?

- how do you deal with aerosol transport (large-scale and small-scale ones)?

- last paragraph of the paper: "Meanwhile a number of developments ...": more complexity is not always the way forward for a climate model. What about going to an on-line description of the aerosols and chemistry, making use of recent developments in the IFS in the MACC and CAMS projects? not to mention the numerical advantages, this would also allow better coherence between EC-Earth and EC-EarthAerChem. Please elaborate on that in the concluding remarks.

**3   Minor remarks**

- line 48: I suggest: "Other EC-Earth CMIP6..."

- line 65: "as follows:"

- line 80: It is "perturbing" for a meteorologist, or for me at least, to refer to IFS here, outside the Numerical Weather Prediction context, and knowing that a number of adaptations have been made to the original IFS. Please indicate more clearly that you call "IFS" the atmospheric component of EC-Earth-AerChem although it difers from the original IFS cy36r4.

- line 97: what is the reference for the stratospheric aerosol radiative properties?

- line 106: from the release 3.6

- line 126: "specific release" instead of "specifically release"?

- line 128: what vertical domain is covered?

- line 149: please specify what "of these particles" refer to

- line 246: no clear whether the three aerosol optical properties are computed on-line at each time step of the radiation scheme (or the aerosol scheme) or whether the calculation of aerosol optical properties has been made just once and look up tables are used. Please clarify.

- line 281: please indicate: 'Thus, the LW absorption in EC-EarthAerChem..."

- line 314: please indicate units

- line 347: please indicate units

- line 390: please clarify what you mean by "calculated from the CMIP6 input data," what are these CMIP6 data?

- line 392: can you explain why this vertical domain was chosen?

- line 444: towards what is the global dust emission tuned?

- line 633: in the SW ERFaci you mean?

- line 707: what are these several reference values?

- line 718: please indicate the offset

- Figure 2 legend: Times series of the global (a) ....and (b)

- Figure 6 legend: please indicate which statistical test has been used

- Figure 7 and 8: more distinct than orange, brown and red colors would help

---

## Referee Comment (RC2) · Anonymous Referee #2 · 17 Mar 2021

The paper by Noije et al describes the new EC-EARTH aerosol and chemistry version. The description of the model used for CMIP6 is very well written, special thanks on that. I definitely think the paper should be published after adding some discussion as suggested below.

General comments:

The description of the aerosol part is quite complete, thanks for that. However, it invites for even more questions. How do you justify the choices made? I believe it would be good to have a discussion paragraph on this. How to test which of the many aerosol

parameterisations is really important for the aerosol ERF?

I find the chemistry description a bit short.

I understand that a more detailed evaluation of the model may be out of scope, but the comparison to EC-Earth3 could be more detailed. That version has quite a different aerosol prescription, so what is the conclusion on having a more complex and more simple aerosol in the same ESM? Do you understand the difference? What are the differences? Why is the ECS higher in EC-Earth3.

The spurious interdecadal variability is striking in the PI control. However, does it really explain the negative GSAT anomaly in the 70s?

The small change in net TOA flux (0.5 W m-2) between historical and hist-piNTCF leads to almost 1 K difference (Figure 7). That would imply a large TCR in that period. Can you comment on possible reasons for that?

Small comments:

L753-755: "surface air and water temperatures may be very different" over the ocean? "a more robust blend of air and water temperatures also from the model" would be better => I wonder if this is more confusing then helpful. I thought the GISTEMP uses SST as a proxy but still pretends that the anomalies reflect SAT also over the ocean. Is Cowtan 2015 suggesting that one should rather use SST from obs and model? Is the result really more meaningful?

L57-58: A little confusing: You simulate methane and ozone, . . . although they are not fully described. please rewrite.

Figure 1: Sea spray factors as a function of SST . not sure this detail is needed. Why this figure and not others on parameterisation details?

---

## Author Comment (AC1) · 14 Apr 2021

We thank referee #1 for his/her thorough and constructive review of our paper. Below we respond to all points raised by the referee.

The authors note this "spurious interdecadal variability" and indicate that it has been seen in other EC-Earth3 configurations described in Doscher et al. GMDD, 2021. They point that the instability is related to the use of NEMO3.6 ocean model and ORCA1 grid. This explanation do not seem fully appropriate as for instance the CMIP6 CNRM climate models (CNRM-CM6-1 and CNRM-ESM2-1), that share the same

ocean model and a comparable ocean grid, show evolutions of the global surface temperature consistent with observations throughout the 20th century.

As noted in Sect. 5, "Parsons et al. (2020) examined interdecadal GSAT variability in pre-industrial control simulations from 39 CMIP6 models. The six models showing the largest variability are EC-Earth3, BCC-CSM2-MR, CNRM-ESM2-1, EC-Earth3-Veg, CNRM-CM6-1 and IPSL-CM6A-LR. Five of these use NEMO3.6 on the ORCA1 grid (EC-Earth3, EC-Earth3-Veg) or extended ORCA1 (eORCA1) grid (CNRM-ESM2-1, CNRM-CM6-1, IPSL-CM6A-LR)." We believe this is a strong indication that the use of NEMO3.6 and the relatively coarse resolution of the ORCA1 grid is part of the explanation for the large internal interdecadal variability simulated by the various standard-resolution configurations of the EC-Earth3 family of models (EC-Earth3, EC-Earth3-Veg and EC-Earth3-AerChem). This doesn't mean that the use of NEMO3.6 on the ORCA1 grid will produce similar behavior irrespective of the details of the NEMO configuration or other model components. In fact, we know that in EC-Earth3 the slow mode of internal variability in GSAT is strongly correlated with the strength of the Atlantic Meridional Overturning Circulation (AMOC), which in turn is driven by deep water formation in the Labrador Sea. Moreover, it has been demonstrated that a weakening of the AMOC corresponds to extended periods with reduced convective activity and high sea ice coverage in the Labrador Sea (Döscher et al., 2021). These processes are state dependent. Thus, the choice of ocean parameters or atmospheric model can have a strong impact on the simulated convection (Koenigk et al., 2020). On the other hand, it is known that the ocean resolution is a critical factor for the deep water formation in the Labrador Sea. The study by Koenigk et al. (2020) showed that in four out of the five models that use NEMO3.6 (HadGEM3-GC31, CMCC-CM2, CNRM-CM6-1, and EC-Earth3P) increasing the ocean resolution from ORCA1 to ORCA025 resulted in increased deep convection in the Labrador Sea. Actually, similar oscillations have been documented for IPSL-CM6A-LR. The mechanisms responsible for the oscillations in this model have recently been investigated in Jiang et al. (Multi-

**GMDD**
centennial variability driven by salinity exchanges between the Atlantic and the Arctic Ocean in a coupled climate model, Journal of Advances in Modeling Earth Systems, 13, e2020MS002366, https://doi.org/10.1029/2020MS002366, 2021) and shown to be associated with freshwater accumulation and release in the Arctic, modulated by the interplay between sea ice and oceanic freshwater export from the Arctic. The EC-Earth consortium has recently started a detailed investigation of the mechanisms underlying the oscillations in the EC-Earth3 model. In the revised manuscript we have clarified the relevant paragraphs and added a reference to the paper by Jiang et al. (2021). We have also extended the pre-industrial control simulation to 500 years and have updated the manuscript accordingly. We have added the time series of the AMOC strength for the pre-industrial simulation in Fig. 2, and included an analysis of the correlation between GSAT and the AMOC strength.

Later on, the authors indicate that "The cooling of the Northern Hemisphere simulated in the 1950s and 1960s may also be caused or enhanced by aerosol effects... Simulations that provide more information on the role of aerosols and their effective radiative forcing contributions are in production".

I would strongly urge the authors to present further analyses on these issues, especially concerning the role of the aerosols.

A more comprehensive analysis of the role played by aerosols would involve estimating the transient and present-day (2014) aerosol effective radiative forcing (ERF). Since the submission of the first version of the manuscript we have completed histSST, histSST-piNTCF, piClim-control and piClim-NTCF, but the aerosol-specific experiments, histSST-piAer and piClim-Aer, are not available yet. We therefore will not be able to present estimates of aerosol ERF in this paper. An in-depth analysis of the transient aerosol ERF and the role played by aerosols in the historical period is planned as part of the EU Horizon 2020 project FORCeS.
Furthermore, such a large spread in the historical simulations raises doubts on evolutions of the other simulations shown. Therefore, analysis of additional members of both the historical simulation, but also the hist-piNTCF simulation, and the scenario simulations ssp370 and ssp370-lowNTCF would be of benefit to the paper. The authors indicate that such additional members are under production.

We have produced four realizations of the historical simulations, and the analysis presented in the first version of the manuscript includes all four of them. For the perturbation and scenario experiments presented in Sect. 4.4 new results have become available, but ensembles are still incomplete. A single realization of each experiment is presented to illustrate the evolution of a few selected variables, including aerosol optical depth. One of the experiments that has been completed since the submission of the first version of the manuscript is the historical simulation with aerosol precursor emissions kept at pre-industrial levels (hist-piAer). We have added this experiment in Figs. 7 and 8. This provides additional information about the role of aerosols in the cooling simulated in the 1950s and 1960s. A more quantitative analysis for these experiments is foreseen once we have filled the matrix of ensemble simulations.

Finally, the impact of the complex aerosol scheme should be better highlighted. Either this impact is important or not, positive or not, in the model behaviour, but this should be made clear in the paper.

The differences between the two configurations are due to a combination of the effects of the different representation of aerosols (and chemistry) and a retuning of the atmosphere. As described in Sect. 3, switching to interactive aerosols and atmospheric chemistry under pre-industrial conditions had a substantial impact on the model's climate, especially in the Northern Hemisphere high-latitude regions where
zonal mean surface air temperatures went up by a few degrees and cold biases were converted into warm biases. The most likely explanation is that correlations between aerosols and clouds on submonthly time scales are better represented. We have clarified this in the revised manuscript. We subsequently reduced these warm biases by adjusting three parameters in the atmospheric GCM. The final configuration is somewhat warmer and exhibits lower natural variability than EC-Earth3 (Sect. 4.1). The main impact is seen in the Northern Hemisphere, where cold biases are substantially reduced in EC-Earth3-AerChem. In the first version of the manuscript we estimated the hemispheric mean pre-industrial temperature biases using a proxy of pre-industrial temperatures estimated from ERA5 and observed hemispheric warming estimates from GISTEMP (Sect. 4.3). In the revised manuscript we have included a more spatially detailed analysis of the pre-industrial temperature bias for both configurations. This indicates more clearly that biases are generally smaller in EC-Earth3-AerChem. Running with interactive aerosols also has an impact on the climate sensitivity; as stated in Sect. 4.2 both the effective equilibrium climate sensitivity (ECS) and the transient climate response (TCR) are reduced in EC-Earth3-AerChem. A better understanding of the impact of the complex aerosol scheme in the historical period would involve a comparison of the transient and present-day (2014) aerosol ERF, preferably including a decomposition into direct radiative forcing, cloud radiative forcing and surface albedo forcing (Ghan, 2013). Such a comparison is beyond the scope of our paper, and can only be made once the required diagnostics from the relevant atmosphere-only simulations for both configurations will be available.

**Comments/questions**

in addition to information of Table 1, it would be of value to add a table with extinction, single- scattering albedo and asymmetry factors, at 550 nm and 80% humidity for instance

The reason we present the refractive index is that it is a material property. Ex-

**GMDD**
tinction (or better mass extinction coefficient which is independent of concentration), single-scattering albedo and asymmetry factor on the other hand depend on the size distribution.

it is not so clear to me how you deal with stratospheric ozone: line 61 you indicate that there is a difference between EC-Earth3 and EC-Earth3- AerChem in "lower-stratospheric ozone": why is that?

and line 383: how do you deal with differences in the tropopause between your model and the CMIP6 ozone data?

The difference between the two configurations is that ozone mixing ratios are prescribed throughout the column in EC-Earth3 (see introduction) while in EC-Earth3-AerChem the nudging scheme described by Williams et al. (2017) is applied (see Sect. 2.3). The nudging scheme only constrains the mixing ratios above certain levels in the stratosphere. The lower level of this domain varies from ~ 45 hPa at low latitude (< 30 degrees), ~ 95 hPa at intermediate latitudes, to ~ 120 hPa at high latitudes (> 66 degrees). In the lower regions of the stratosphere no nudging is applied, and the mixing ratios are determined by transport and to lesser extent chemistry. In this region mixing ratios are influenced by the CMIP6 ozone data only by downward transport from upper levels. Consequently, the location of the tropopause is fully consistent with the ozone concentrations simulated in the model.

line 215: please indicate how the wet deposition of aerosol is performed, from convective precipitation and from stratiform precipitation, considering 3D precipitation fields from IFS or...?

We have clarified in the revision that the scavenging by precipitation formation in convective clouds and the below-cloud scavenging by stratiform precipitation are both calculated based on surface precipitation fields from IFS, as in the standalone version

GMDD
of TM5. We have included a statement explaining that the convective scavenging is included in the convective mass transport operator, as in the model version documented by van Noije et al. (2014). For details on the procedure to calculate the vertical profile of stratiform precipitation we have added a reference to the paper by de Bruine et al. (2018).

line 338: could you detail why you use fixed values of CDNC here?

The calculation of photolysis rates in TM5 follows the description in Williams et al. (2017). As described in the text it used fixed values of CDNC over land and ocean, which originate from the original IFS model. When introducing the Abdul-Razzak and Ghan scheme for EC-Earth, we didn't feed the resulting CDNC values through to TM5. This could be improved in a future version of the model.

line 423: it would be interesting to have a Table summarizing emitted quantities

Information on emissions, together with burdens and lifetimes of the various aerosol components, for both TM5 and a nudged version of EC-Earth3-AerChem are provided in the paper by Gliß et al. (2021). We will include an additional reference to this paper.

*line 502: I could not find in Dentener et al. 2005 recommendation for emission of SO4. Please provide further details*

One of the footnotes to Table 1 in the paper by Dentener et al. (2006) specifies that 2.5 % of sulfur should be emitted as sulfate.

line 544: you use a fixed number of layers, independently of the tropopause height?

GMDD
Yes, for ease of implementation we use a fixed number of 23 levels, which under all circumstances cover the whole troposphere.

line 552: have you done any testing for 3h exchange of fields?

We have experimented with this when developing earlier versions of the model. We haven't explored this further for EC-Earth3-AerChem. As noted in the text, this would lead to a substantial decline in performance.

*line 608: please indicate names of such regions you observed such a discrepancy in particle number concentration*

We have indicated the regions in the text.

line 620: it is more common to nudge winds. Do you have a reference for such a configuration?

Indeed we did nudge winds (vorticity and divergence) and not temperature. Thanks for spotting, we have corrected the text.

line 780: a usual climatology covers a 30-year period. Why did you restrict it to a 20-year period?

The aim is to estimate the bias at the end of the historical period. For this we think a period of 20 years is a reasonable compromise. A 30-year climatology would be more strongly affected by the drop in Northern Hemisphere temperatures simulated during the 1950s and 1960s.
line 784: biases over Antarctica seem to be larger than 6K, please specify the maximum biases

We have added these in the text.

Table 6: could you add fields transferred from IFS to TM5?

As this is quite a long list, we refer to the list given in van Noije et al. (2014). All updates are specified in detail in lines 529 to 535 of the original manuscript.

how do you deal with aerosol transport (large-scale and small-scale ones)?

Aerosol tracers in TM5 are transported by advection, cumulus convection, vertical diffusion, and sedimentation. Details are given in van Noije et al. (2014). We have included a statement to clarify this.

last paragraph of the paper: "Meanwhile a number of developments ...": more complexity is not always the way forward for a climate model. What about going to an on-line description of the aerosols and chemistry, making use of recent developments in the IFS in the MACC and CAMS projects? not to mention the numerical advantages, this would also allow better coherence between EC-Earth and EC-EarthAerChem. Please elaborate on that in the concluding remarks.

In the concluding paragraph we listed some recent developments and concrete plans for improving EC-Earth3-AerChem post CMIP6. In parallel the EC-Earth community has started the development of EC-Earth4, which will be based on OpenIFS. The development of an OpenIFS version with integrated description of aerosols and chemistry is in progress. In the revised manuscript we have mentioned this parallel development in the conclusions.
Minor remarks line 48: I suggest: "Other EC-Earth CMIP6..."

Okay, we have changed the text.

line 65: "as follows:"

Okay

line 80: It is "perturbing" for a meteorologist, or for me at least, to refer to IFS here, outside the Numerical Weather Prediction context, and knowing that a number of adaptations have been made to the original IFS. Please indicate more clearly that you call "IFS" the atmospheric component of EC-Earth-AerChem although it differs from the original IFS cy36r4.

Point taken. We have changed "The time step applied in IFS is 45 min." to "Following ECMWF recommendations, the model time step for this resolution is set to 45 min." In line 95 we have changed "the CMIP6 forcings prescribed in IFS" to "the CMIP6 forcings prescribed in the modified IFS model". At the end of the section describing the atmospheric GCM, we now explicitly state the following: "In the remainder of this paper we will refer to the atmospheric GCM of EC-Earth simply as 'IFS'." Finally, in the concluding section, we have changed: "It uses a coupling between IFS and TM5" to "It uses a coupling between an IFS based atmospheric GCM and the TM5 atmospheric chemistry and transport model".

line 97: what is the reference for the stratospheric aerosol radiative properties?

No proper reference exists for this data set, as far as we know. Else we would
have included it.

line 106: from the release 3.6

Okay, we have changed "the release" to "release 3.6".

line 126: "specific release" instead of "specifically release"?

We considered the reviewer's suggestion, but we believe "specifically" is correct in this case.

line 128: what vertical domain is covered?

Details about the TM5 grid are given in Sect. 2.5, where it is explained that the top layer has a full-level pressure of  $\sim 0.1$  hPa.

line 149: please specify what "of these particles" refer to

Here "these particles" refers to "the soluble accumulation-mode particles" in the preceding sentence. We have clarified this by changing the text to "the particles in this mode".

line 246: no clear whether the three aerosol optical properties are computed online at each time step of the radiation scheme (or the aerosol scheme) or whether the calculation of aerosol optical properties has been made just once and look up tables are used. Please clarify.

We refer to the paper by van Noije et al. (2014), where details are given: "The optical properties of the aerosol mixtures are calculated as a function of wavelength
based on Mie theory, using a look-up table (Aan de Brugh et al., 2011; Aan de Brugh, 2013)." We have clarified this.

line 281: please indicate: 'Thus, the LW absorption in EC-EarthAerChem..."

Okay, we have changed the text to "Thus, the LW absorption in our model ....".

line 314: please indicate units line 347: please indicate units

We don't think this is needed here, because we don't provide values for any of the quantities appearing in these formulas. Details are given in the referenced original literature.

line 390: please clarify what you mean by "calculated from the CMIP6 input data," what are these CMIP6 data?

We have clarified this.

line 392: can you explain why this vertical domain was chosen?

This was the outcome of tests performed with the offline TM5 model (e.g. Williams et al., 2017). The domain height has been increased from ~ 2 km in the study by Bândă et al. (2014) to ~ 5 km in the present version of the model. In view of the long lifetime of methane, we believe the results will be relatively insensitive to the chosen domain height.

line 444: towards what is the global dust emission tuned?

GMDD
The correction factor for the friction velocity was determined by comparing the aerosol optical depth from offline TM5 simulations driven by ERA-Interim against satellite observations from MODIS. We have later evaluated the model with this parameter setting against more extensive observational data sets of (dust) optical depth and dust surface concentrations. These and other studies, such as the recent AeroCom intercomparison study performed by Gliß et al. (2021) and the study by Checa-Garcia et al. (2021), indicate that the model performs reasonably well in simulating the atmospheric dust cycle. The same value of the correction factor was adopted in EC-Earth3-AerChem. As described in Sect. 3, we have verified that this results in a global total amount of dust emissions close to that obtained with the standalone TM5 model.

line 633: in the SW ERFaci you mean?

Indeed, it's a change in the cloud forcing in response to a perturbation of aerosol emissions. We have clarified this.

line 707: what are these several reference values?

The reference values are the annual values from the linear fit.

line 718: please indicate the offset

As noted in Sect. 4.1, the pre-industrial simulation shows no significant drift in the TOA flux. Hence, the offset will be very close to the mean value given in Fig. 2.

Figure 2 legend: Times series of the global (a) ....and (b)

Okay, we have added "global" before "net TOA flux".
Figure 6 legend: please indicate which statistical test has been used

Okay, we have clarified this.

Figure 7 and 8: more distinct than orange, brown and red colors would help

To make these plots we have selected the default qualitative colour palette from the Seaborn data visualization library in Python. We think this is fit for purpose. Anyway, the colours for the scenario experiments have changed because of the inclusion of hist-piAer in these figures.

---

## Author Comment (AC2) · 14 Apr 2021

We thank referee #2 for his/her positive and helpful review of our paper. Below we respond to all points raised by the referee.

*The description of the aerosol part is quite complete, thanks for that. However, it invites for even more questions. How do you justify the choices made? I believe it would be good to have a discussion paragraph on this. How to test which of the many aerosol parameterisations is really important for the aerosol ERF?*

That's a fair point. When developing the model, we have critically assessed various aspects of the models. As part of this exercise parameter settings were revised in accordance with recent literature, and sensitivity simulations were performed to test the outcome against observations, mainly aerosol optical depth. Later, more in-depth evaluations have been performed using a variety of observational data sets. Recent examples are the study by Bergman et al. (2021), which focuses on secondary organic aerosols and new particle formation, and the model intercomparison studies by Gliß et al. (2021) and Checa-Garcia et al. (2021). The latter studies show that the model is among the best ones when it comes to aerosol metrics. The main purpose of this paper, however, is not to justify the choices made but to document the model and present some first results from our CMIP6 simulations. In Sect. 3 we give one example of how an adjustment in the treatment of emissions of carbonaceous aerosols has led to a reduction of the aerosol ERF. In general, perturbed parameter experiments (PPEs) can be used to test the impact on the aerosol ERF. Such experiments have been proposed within AeroCom and are currently being discussed as part of the EU Horizon 2020 project FORCeS.

*I find the chemistry description a bit short.*

This is because the chemistry scheme is largely the same as in the TM5 model version described by Williams et al. (2017), which documents it in great detail. We didn't want to repeat that information in our paper. We therefore described the general characteristics and specified the changes and new features in more detail.

*I understand that a more detailed evaluation of the model may be out of scope, but the comparison to EC-Earth3 could be more detailed. That version has quite a different aerosol prescription, so what is the conclusion on having a more complex and more simple aerosol in the same ESM? Do you understand the difference? What are the differences? Why is the ECS higher in EC-Earth3.*

The differences between the two configurations are due to a combination of the effects of the different representation of aerosols (and chemistry) and a retuning of the atmosphere. As described in Sect. 3, switching to interactive aerosols and atmospheric chemistry under pre-industrial conditions had a substantial impact on the model's climate, especially in the Northern Hemisphere high-latitude regions where zonal mean surface air temperatures went up by a few degrees and cold biases were converted into warm biases. The most likely explanation is that correlations between aerosols and clouds on submonthly time scales are better represented. We have clarified this in the revised manuscript. We subsequently reduced these warm biases by adjusting three parameters in the atmospheric GCM. The final configuration is somewhat warmer and exhibits lower natural variability than EC-Earth3 (Sect. 4.1). The main impact is seen in the Northern Hemisphere, where cold biases are substantially reduced in EC-Earth3-AerChem. In the first version of the manuscript we estimated the hemispheric mean pre-industrial temperature biases using a proxy of pre-industrial temperatures estimated from ERA5 and observed hemispheric warming estimates from GISTEMP (Sect. 4.3). In the revised manuscript we have included a more spatially detailed analysis of the pre-industrial temperature bias for both configurations. This indicates more clearly that biases are generally smaller in EC-Earth3-AerChem. We have also extended the pre-industrial control simulation to 500 years and have updated the manuscript accordingly. A better understanding of the impact of the complex aerosol scheme in the historical period would involve a comparison of the transient and present-day (2014) aerosol ERF, preferably including a decomposition into direct radiative forcing, cloud radiative forcing and surface albedo forcing (Ghan, 2013). Such a comparison is beyond the scope of our paper, and can only be made once the required diagnostics from the relevant atmosphere-only simulations for both configurations will be available (see our answer to the next point). We currently do not have an explanation for the difference in effective equilibrium climate sensitivity and transient climate response between the two configurations.

*The spurious interdecadal variability is striking in the PI control. However, does it really explain the negative GSAT anomaly in the 70s?*

In the first version of the manuscript we wrote: "The cooling of the Northern Hemisphere simulated in the 1950s and 1960s may also be caused or enhanced by aerosol effects. To what extent this is the case needs further investigation. Simulations that provide more information on the role of aerosols and their effective radiative forcing contributions are in production." More output has become available since we submitted the first version of our manuscript. In particular, we have completed the historical simulation with aerosol precursor emissions kept at pre-industrial levels (hist-piAer). We have added this experiment in Figs. 7 and 8. This provides additional information about the role of aerosols in the cooling simulated in the 1950s and 1960s. A more comprehensive analysis of the role played by aerosols would involve estimating the transient and present-day (2014) aerosol effective radiative forcing (ERF). Since the submission of the first version of the manuscript we have completed histSST, histSST-piNTCF, piClim-control and piClim-NTCF, but the aerosol-specific experiments, histSST-piAer and piClim-Aer, are not available yet. We therefore will not be able to present estimates of aerosol ERF in this paper. An in-depth analysis of the transient aerosol ERF and the role played by aerosols in the historical period is planned as part of the EU Horizon 2020 project FORCeS.

*The small change in net TOA flux (0.5 W m-2) between historical and hist-piNTCF leads to almost 1 K difference (Figure 7). That would imply a large TCR in that period. Can you comment on possible reasons for that?*

To make such an inference, one needs information about the effective radiative forcing by NTCFs. This can be estimated by differencing histSST and histSST-piNTCF. The difference in the TOA fluxes shown in Fig. 7 doesn't provide the ERF, as it is

modified by the response of SSTs and sea ice concentrations.

*Small comments:*
*L753-755: "surface air and water temperatures may be very different" over the ocean?*

Yes, according to the GISTEMP website: "SATs and SSTs may be very different (since air warms and cools much faster than water)".

*"a more robust blend of air and water temperatures also from the model" would be better => I wonder if this is more confusing than helpful. I thought the GISTEMP uses SST as a proxy but still pretends that the anomalies reflect SAT also over the ocean.*

That's correct.

*Is Cowtan 2015 suggesting that one should rather use SST from obs and model?*

Correct, that would be the most consistent way to make the comparison.

*Is the result really more meaningful?*

For the purpose of our paper, it wouldn't make much of a difference. We only included this sentence to justify the method adopted for comparing our model with the observations.

*L57-58: A little confusing: You simulate methane and ozone, ... although they are not fully described. please rewrite.*

Okay, we have clarified this sentence.

*Figure 1: Sea spray factors as a function of SST . not sure this detail is needed. Why this figure and not others on parameterisation details?*

The reason the temperature dependence of sea spray formation is described in detail is that the scale factors applied in our model are not exactly the same as the expression from Salter et al. (2015). Moreover, these are single-variable functional relations that can easily be shown in a graph.

---

## Author Response (AR1)

We have prepared a revised version of our manuscript and have thereby responded to all comments made by the reviewers, according to our author comments published on 14 April. For reference, we provide a track-changes word document of the revised manuscript. In particular, the results from the pre-industrial control simulation have been updated and are now based on the extended 500-year simulation. We have expanded the analysis of the model's pre-industrial climate and interdecadal variability, and have included a comparison with the standard EC-Earth3 configuration, as requested by the reviewers. Moreover, results from hist-piAer have been added, and we now compare the temperature evolution in hist-piNTCF and hist-piAer to the range provided by the four-member historical ensemble. In addition, we more clearly distinguish between uncertainty ranges (e.g. in mean quantities) and ranges given in terms of standard deviations (in interannual quantities or across ensemble members).

---

## Author Response (AR2)

Dear topical editor, dear Fiona,

We are grateful that our manuscript has been accepted for publication in Geoscientific Model Developments.

We have prepared all files required for the final submission. Herewith we would like to point out that in our final manuscript we have included two minor corrections and updated one of the references:

1) In table 2 we have corrected the scavenging fraction for the scavenging of bulk aerosols in stratiform ice clouds from 0.0 to 0.06. This was a typo in the original manuscript, which we only recently discovered.

2) In line 236, we included "and stratiform". The sentence now reads "As in earlier versions of the model, the scavenging by precipitation formation in convective **and stratiform** clouds and the below-cloud scavenging by stratiform precipitation are calculated in TM5 using surface precipitation fields received from IFS."

3) We updated the reference to the paper in Atmospheric Chemistry and Physics by Checa-Garcia et al., which was recently accepted.

Yours Sincerely,
Twan van Noije